# ORCHESTRATED VALUE MAPPING
# FOR REINFORCEMENT LEARNING

**Mehdi Fatemi**
Microsoft Research
Montréal, Canada
mehdi.fatemi@microsoft.com

**Arash Tavakoli**
Max Planck Institute for Intelligent Systems
Tübingen, Germany
arash.tavakoli@tuebingen.mpg.de

## ABSTRACT

We present a general convergent class of reinforcement learning algorithms that is founded on two distinct principles: (1) mapping value estimates to a different space using arbitrary functions from a broad class, and (2) linearly decomposing the reward signal into multiple channels. The first principle enables incorporating specific properties into the value estimator that can enhance learning. The second principle, on the other hand, allows for the value function to be represented as a composition of multiple utility functions. This can be leveraged for various purposes, e.g. dealing with highly varying reward scales, incorporating a priori knowledge about the sources of reward, and ensemble learning. Combining the two principles yields a general blueprint for instantiating convergent algorithms by orchestrating diverse mapping functions over multiple reward channels. This blueprint generalizes and subsumes algorithms such as Q-Learning, Log Q-Learning, and Q-Decomposition. In addition, our convergence proof for this general class relaxes certain required assumptions in some of these algorithms. Based on our theory, we discuss several interesting configurations as special cases. Finally, to illustrate the potential of the design space that our theory opens up, we instantiate a particular algorithm and evaluate its performance on the Atari suite.

## 1 INTRODUCTION

The chief goal of reinforcement learning (RL) algorithms is to maximize the expected return (or the value function) from each state (Szepesvári, 2010; Sutton & Barto, 2018). For decades, many algorithms have been proposed to compute target value functions either as their main goal (critic-only algorithms) or as a means to help the policy search process (actor-critic algorithms). However, when the environment features certain characteristics, learning the underlying value function can become very challenging. Examples include environments where rewards are dense in some parts of the state space but very sparse in other parts, or where the scale of rewards varies drastically. In the Atari 2600 game of Ms. Pac-Man, for instance, the reward can vary from 10 (for small pellets) to as large as 5000 (for moving bananas). In other games such as Tennis, acting randomly leads to frequent negative rewards and losing the game. Then, once the agent learns to capture the ball, it can avoid incurring such penalties. However, it may still take a very long time before the agent scores a point and experiences a positive reward. Such learning scenarios, for one reason or another, have proved challenging for the conventional RL algorithms.

One issue that can arise due to such environmental challenges is having highly non-uniform action gaps across the state space.[1] In a recent study, van Seijen et al. (2019) showed promising results by simply mapping the value estimates to a logarithmic space and adding important algorithmic components to guarantee convergence under standard conditions. While this construction addresses the problem of non-uniform action gaps and enables using lower discount factors, it further opens a new direction for improving the learning performance: estimate the value function in a different space that admits better properties compared to the original space. This interesting view naturally raises theoretical questions about the required properties of the mapping functions, and whether the guarantees of convergence would carry over from the basis algorithm under this new construction.

---

[1] Action gap refers to the value difference between optimal and second best actions (Farahmand, 2011).

One loosely related topic is that of nonlinear Bellman equations. In the canonical formulation of Bellman equations (Bellman, 1954; 1957), they are limited in their modeling power to cumulative rewards that are discounted exponentially. However, one may go beyond this basis and redefine the Bellman equations in a general nonlinear manner. In particular, van Hasselt et al. (2019) showed that many such Bellman operators are still contraction mappings and thus the resulting algorithms are reasonable and inherit many beneficial properties of their linear counterparts. Nevertheless, the application of such algorithms is still unclear since the fixed point does not have a direct connection to the concept of return. In this paper we do not consider nonlinear Bellman equations.

Continuing with the first line of thought, a natural extension is to employ multiple mapping functions concurrently in an ensemble, allowing each to contribute their own benefits. This can be viewed as a form of separation of concerns (van Seijen et al., 2016). Ideally, we may want to dynamically modify the influence of different mappings as the learning advances. For example, the agent could start with mappings that facilitate learning on sparse rewards. Then, as it learns to collect more rewards, the mapping function can be gradually adapted to better support learning on denser rewards. Moreover, there may be several sources of reward with specific characteristics (e.g. sparse positive rewards but dense negative ones), in which case using a different mapping to deal with each reward channel could prove beneficial.

Building upon these ideas, this paper presents a general class of algorithms based on the combination of two distinct principles: value mapping and linear reward decomposition. Specifically, we present a broad class of mapping functions that inherit the convergence properties of the basis algorithm. We further show that such mappings can be orchestrated through linear reward decomposition, proving convergence for the complete class of resulting algorithms. The outcome is a *blueprint* for building new convergent algorithms as instances. We conceptually discuss several interesting configurations, and experimentally validate one particular instance on the Atari 2600 suite.

## 2 VALUE MAPPING

We consider the standard reinforcement learning problem which is commonly modeled as a Markov decision process (MDP; Puterman (1994)) $\mathcal{M} = (\mathcal{S}, \mathcal{A}, P, R, P_0, \gamma)$, where $\mathcal{S}$ and $\mathcal{A}$ are the discrete sets of states and actions, $P(s'|s,a) \doteq \mathbb{P}[s_{t+1} = s' \mid s_t = s, a_t = a]$ is the state-transition distribution, $R(r|s,a,s') \doteq \mathbb{P}[r_t = r \mid s_t = s, a_t = a, s_{t+1} = s']$ (where we assume $r \in [r_{\min}, r_{\max}]$) is the reward distribution, $P_0(s) \doteq \mathbb{P}[s_0 = s]$ is the initial-state distribution, and $\gamma \in [0, 1]$ is the discount factor. A policy $\pi(a|s) \doteq \mathbb{P}[a_t = a \mid s_t = s]$ defines how an action is selected in a given state. Thus, selecting actions according to a stationary policy generally results in a stochastic trajectory. The discounted sum of rewards over the trajectory induces a random variable called the *return*. We assume that all returns are finite and bounded. The state-action value function $Q_\pi(s,a)$ evaluates the expected return of taking action $a$ at state $s$ and following policy $\pi$ thereafter. The optimal value function is defined as $Q^*(s,a) \doteq \max_\pi Q_\pi(s,a)$, which gives the maximum expected return of all trajectories starting from the state-action pair $(s,a)$. Similarly, an optimal policy is defined as $\pi^*(a|s) \in \arg\max_\pi Q_\pi(s,a)$. The optimal value function is unique (Bertsekas & Tsitsiklis, 1996) and can be found, e.g., as the fixed point of the Q-Learning algorithm (Watkins, 1989; Watkins & Dayan, 1992) which assumes the following update:

$$Q_{t+1}(s_t, a_t) \leftarrow (1 - \alpha_t)Q_t(s_t, a_t) + \alpha_t\big(r_t + \gamma \max_{a'} Q_t(s_{t+1}, a')\big), \tag{1}$$

where $\alpha_t$ is a positive learning rate at time $t$. Our goal is to map $Q$ to a different space and perform the update in that space instead, so that the learning process can benefit from the properties of the mapping space.

We define a function $f$ that maps the value function to some new space. In particular, we consider the following assumptions:

**Assumption 1** *The function $f(x)$ is a bijection (either strictly increasing or strictly decreasing) for all $x$ in the given domain $\mathcal{D} = [c_1, c_2] \subseteq \mathbb{R}$.*

**Assumption 2** *The function $f(x)$ holds the following properties for all $x$ in the given domain $\mathcal{D} = [c_1, c_2] \subseteq \mathbb{R}$:*

   *1. $f$ is continuous on $[c_1, c_2]$ and differentiable on $(c_1, c_2)$;*

2. $|f'(x)| \in [\delta_1, \delta_2]$ *for* $x \in (c_1, c_2)$*, with* $0 < \delta_1 < \delta_2 < \infty$*;*

3. *$f$ is either of semi-convex or semi-concave.*

We next use $f$ to map the value function, $Q(s, a)$, to its transformed version, namely

$$\widetilde{Q}(s, a) \doteq f\big(Q(s, a)\big). \tag{2}$$

Assumption 1 implies that $f$ is invertible and, as such, $Q(s, a)$ is uniquely computable from $\widetilde{Q}(s, a)$ by means of the inverse function $f^{-1}$. Of note, this assumption also implies that $f$ preserves the ordering in $x$; however, it inverts the ordering direction if $f$ is decreasing. Assumption 2 imposes further restrictions on $f$, but still leaves a broad class of mapping functions to consider. Throughout the paper, we use *tilde* to denote a "mapped" function or variable, while the mapping $f$ is understandable from the context (otherwise it is explicitly said).

## 2.1 Base Algorithm

If mapped value estimates were naively placed in a Q-Learning style algorithm, the algorithm would fail to converge to the optimal values in stochastic environments. More formally, in the tabular case, an update of the form (cf. Equation 1)

$$\widetilde{Q}_{t+1}(s_t, a_t) \leftarrow (1 - \alpha_t)\widetilde{Q}_t(s_t, a_t) + \alpha_t f\left(r_t + \gamma \max_{a'} f^{-1}\big(\widetilde{Q}_t(s_{t+1}, a')\big)\right) \tag{3}$$

converges[2] to the fixed point $\widetilde{Q}^{\odot}(s, a)$ that satisfies

$$\widetilde{Q}^{\odot}(s, a) = \mathbb{E}_{s' \sim P(\cdot|s,a),\, r \sim R(\cdot|s,a,s')} \left[ f\left( r + \gamma \max_{a'} f^{-1}\big(\widetilde{Q}^{\odot}(s', a')\big)\right)\right]. \tag{4}$$

Let us define the notation $Q^{\odot}(s, a) \doteq f^{-1}\big(\widetilde{Q}^{\odot}(s, a)\big)$. If $f$ is a semi-convex bijection, $f^{-1}$ will be semi-concave and Equation 4 deduces

$$
\begin{aligned}
Q^{\odot}(s, a) &\doteq f^{-1}\big(\widetilde{Q}^{\odot}(s, a)\big) \\
&= f^{-1}\left(\mathbb{E}_{s' \sim P(\cdot|s,a),\, r \sim R(\cdot|s,a,s')}\left[f\left(r + \gamma \max_{a'} Q^{\odot}(s', a')\right)\right]\right) \\
&\geq \mathbb{E}_{s' \sim P(\cdot|s,a),\, r \sim R(\cdot|s,a,s')}\left[f^{-1}\left(f\left(r + \gamma \max_{a'} Q^{\odot}(s', a')\right)\right)\right] \\
&= \mathbb{E}_{s' \sim P(\cdot|s,a),\, r \sim R(\cdot|s,a,s')}\left[r + \gamma \max_{a'} Q^{\odot}(s', a')\right],
\end{aligned} \tag{5}
$$

where the third line follows Jensen's inequality. Comparing Equation 5 with the Bellman optimality equation in the regular space, i.e. $Q^*(s, a) = \mathbb{E}_{s',r \sim P,R}[r + \gamma \max_{a'} Q^*(s', a')]$, we conclude that the value function to which the update rule (3) converges overestimates Bellman's backup. Similarly, if $f$ is a semi-concave function, then $Q^{\odot}(s, a)$ underestimates Bellman's backup. Either way, it follows that the learned value function deviates from the optimal one. Furthermore, the Jensen's gap at a given state $s$ — the difference between the left-hand and right-hand sides of Equation 5 — depends on the action $a$ because the expectation operator depends on $a$. That is, at a given state $s$, the deviation of $Q^{\odot}(s, a)$ from $Q^*(s, a)$ is not a fixed-value shift and can vary for various actions. Hence, the greedy policy w.r.t. (with respect to) $Q^{\odot}(s, \cdot)$ may not preserve ordering and it may not be an optimal policy either.

In an effort to address this problem in the spacial case of $f$ being a logarithmic function, van Seijen et al. (2019) observed that in the algorithm described by Equation 3, the learning rate $\alpha_t$ generally conflates two forms of averaging: (i) averaging of stochastic update targets due to environment stochasticity (happens in the regular space), and (ii) averaging over different states and actions (happens in the $f$'s mapping space). To this end, they proposed to algorithmically disentangle the two and showed that such a separation will lift the Jensen's gap if the learning rate for averaging in the regular space decays to zero fast enough.

Building from Log Q-Learning (van Seijen et al., 2019), we define the base algorithm as follows: at each time $t$, the algorithm receives $\widetilde{Q}_t(s, a)$ and a transition quadruple $(s, a, r, s')$, and outputs $\widetilde{Q}_{t+1}(s, a)$, which then yields $Q_{t+1}(s, a) \doteq f^{-1}\big(\widetilde{Q}_{t+1}(s, a)\big)$. The steps are listed below:

---

[2]The convergence follows from stochastic approximation theory with the additional steps to show by induction that $\widetilde{Q}$ remains bounded and then the corresponding operator is a contraction mapping.

$$Q_t(s,a) := f^{-1}\left(\widetilde{Q}_t(s,a)\right) \tag{6}$$

$$\tilde{a}' := \arg\max_{a'}\left(Q_t(s',a')\right) \tag{7}$$

$$U_t := r + \gamma f^{-1}\left(\widetilde{Q}_t(s',\tilde{a}')\right) \tag{8}$$

$$\hat{U}_t := f^{-1}\left(\widetilde{Q}_t(s,a)\right) + \beta_{reg,t}\left(U_t - f^{-1}\left(\widetilde{Q}_t(s,a)\right)\right) \tag{9}$$

$$\widetilde{Q}_{t+1}(s,a) := \widetilde{Q}_t(s,a) + \beta_{f,t}\left(f(\hat{U}_t) - \widetilde{Q}_t(s,a)\right) \tag{10}$$

Here, the mapping $f$ is any function that satisfies Assumptions 1 and 2. Remark that similarly to the Log Q-Learning algorithm, Equations 9 and 10 have decoupled averaging of stochastic update targets from that over different states and actions.

## 3 REWARD DECOMPOSITION

Reward decomposition can be seen as a generic way to facilitate (i) systematic use of environmental inductive biases in terms of known reward sources, and (ii) action selection as well as value-function updates in terms of communication between an arbitrator and several subagents, thus assembling several subagents to collectively solve a task. Both directions provide broad avenues for research and have been visited in various contexts. Russell & Zimdars (2003) introduced an algorithm called Q-Decomposition with the goal of extending beyond the "monolithic" view of RL. They studied the case of additive reward channels, where the reward signal can be written as the sum of several reward channels. They observed, however, that using Q-Learning to learn the corresponding $Q$ function of each channel will lead to a non-optimal policy (they showed it through a counterexample). Hence, they used a Sarsa-like update w.r.t. the action that maximizes the arbitrator's value. Laroche et al. (2017) provided a formal analysis of the problem, called the *attractor* phenomenon, and studied a number of variations to Q-Decomposition. On a related topic but with a different goal, Sutton et al. (2011) introduced the Horde architecture, which consists of a large number of "demons" that learn in parallel via off-policy learning. Each demon estimates a separate value function based on its own target policy and (pseudo) reward function, which can be seen as a decomposition of the original reward in addition to auxiliary ones. van Seijen et al. (2017) built on these ideas and presented hybrid reward architecture (HRA) to decompose the reward and learn their corresponding value functions in parallel, under *mean bootstrapping*. They further illustrated significant results on domains with many independent sources of reward, such as the Atari 2600 game of Ms. Pac-Man.

Besides utilizing distinct environmental reward sources, reward decomposition can also be used as a technically-sound algorithmic machinery. For example, reward decomposition can enable utilization of a specific mapping that has a limited domain. In the Log Q-Learning algorithm, for example, the $\log(\cdot)$ function cannot be directly used on non-positive values. Thus, the reward is decomposed such that two utility functions $\widetilde{Q}^+$ and $\widetilde{Q}^-$ are learned for when the reward is non-negative or negative, respectively. Then the value is given by $Q(s,a) = \exp\left(\widetilde{Q}^+(s,a)\right) - \exp\left(\widetilde{Q}^-(s,a)\right)$. The learning process of each of $\widetilde{Q}^+$ and $\widetilde{Q}^-$ bootstraps towards their corresponding value estimate at the next state with an action that is the $\arg\max$ of the actual $Q$, rather than that of $\widetilde{Q}^+$ and $\widetilde{Q}^-$ individually.

We generalize this idea to incorporate arbitrary decompositions, beyond only two channels. To be specific, we are interested in linear decompositions of the reward function into $L$ separate channels $r^{(j)}$, for $j = 1 \ldots L$, in the following way:

$$r := \sum_{j=1}^{L} \lambda_j r^{(j)}, \tag{11}$$

with $\lambda_j \in \mathbb{R}$. The channel functions $r^{(j)}$ map the original reward into some new space in such a way that their weighted sum recovers the original reward. Clearly, the case of $L = 1$ and $\lambda_1 = 1$ would retrieve the standard scenario with no decomposition. In order to provide the update, expanding from Log Q-Learning, we define $\widetilde{Q}^{(j)}$ for $j = 1 \ldots L$, corresponding to the above reward channels,

and construct the actual value function $Q$ using the following:

$$Q_t(s,a) := \sum_{j=1}^{L} \lambda_j f_j^{-1}\left(\widetilde{Q}_t^{(j)}(s,a)\right).$$ (12)

We explicitly allow the mapping functions, $f_j$, to be different for each channel. That is, each reward channel can have a different mapping and each $\widetilde{Q}^{(j)}$ is learned separately under its own mapping. Before discussing how the algorithm is updated with the new channels, we present a number of interesting examples of how Equation 11 can be deployed.

As the first example, we can recover the original Log Q-Learning reward decomposition by considering $L = 2$, $\lambda_1 = +1$, $\lambda_2 = -1$, and the following channels:

$$r_t^{(1)} := \begin{cases} r_t & \text{if } r_t \geq 0 \\ 0 & \text{otherwise} \end{cases} \quad ; \quad r_t^{(2)} := \begin{cases} |r_t| & \text{if } r_t < 0 \\ 0 & \text{otherwise} \end{cases}$$ (13)

Notice that the original reward is retrieved via $r_t = r_t^{(1)} - r_t^{(2)}$. This decomposition allows for using a mapping with only positive domain, such as the logarithmic function. This is an example of using reward decomposition to ensure that values do not cross the domain of mapping function $f$.

In the second example, we consider different magnifications for different sources of reward in the environment so as to make the channels scale similarly. The Atari 2600 game of Ms. Pac-Man is an example which includes rewards with three orders of magnitude difference in size. We may therefore use distinct channels according to the size-range of rewards. To be concrete, let $r \in [0, 100]$ and consider the following two configurations for decomposition (can also be extended to other ranges).

---

**Configuration 1:**

$$\lambda_1 = 1, \quad \lambda_2 = 10, \quad \lambda_3 = 100$$

$$r_t^{(1)} := \begin{cases} r_t & \text{if } r_t \in [0,1] \\ 0 & r_t > 1 \end{cases}$$

$$r_t^{(2)} := \begin{cases} 0 & \text{if } r_t \leq 1 \\ 0.1 r_t & \text{if } r_t \in (1,10] \\ 0 & r_t > 10 \end{cases}$$

$$r_t^{(3)} := \begin{cases} 0 & \text{if } r_t \leq 10 \\ 0.01 r_t & \text{if } r_t \in (10,100] \end{cases}$$

**Configuration 2:**

$$\lambda_1 = 1, \quad \lambda_2 = 9, \quad \lambda_3 = 90$$

$$r_t^{(1)} := \begin{cases} r_t & \text{if } r_t \in [0,1] \\ 1 & r_t > 1 \end{cases}$$

$$r_t^{(2)} := \begin{cases} 0 & \text{if } r_t \leq 1 \\ (r_t-1)/9 & \text{if } r_t \in (1,10] \\ 1 & r_t > 10 \end{cases}$$

$$r_t^{(3)} := \begin{cases} 0 & \text{if } r_t \leq 10 \\ (r_t-10)/90 & \text{if } r_t \in (10,100] \end{cases}$$

---

Each of the above configurations presents certain characteristics. Configuration 1 gives a scheme where, at each time step, at most one channel is non-zero. Remark, however, that each channel will be non-zero with less frequency compared to the original reward signal, since rewards get assigned to different channels depending on their size. On the other hands, Configuration 2 keeps each channel to act as if there is a reward clipping at its upper bound, while each channel does not see rewards below its lower bound. As a result, Configuration 2 fully preserves the reward density at the first channel and presents a better density for higher channels compared to Configuration 1. However, the number of active channels depends on the reward size and can be larger than one. Importantly, the magnitude of reward for all channels always remains in $[0, 1]$ in both configurations, which could be a desirable property. The final point to be careful about in using these configurations is that the large weight of higher channels significantly amplifies their corresponding value estimates. Hence, even a small estimation error at higher channels can overshadow the lower ones.

Over and above the cases we have presented so far, reward decomposition enables an algorithmic machinery in order to utilize various mappings concurrently in an ensemble. In the simplest case, we note that in Equation 11 by construction we can always write

$$r^{(j)} := r \quad \text{and} \quad \sum_{j=1}^{L} \lambda_j := 1.$$ (14)

That is, the channels are merely the original reward with arbitrary weights that should sum to one. We can then use arbitrary functions $f_j$ for different channels and build the value function as presented in Equation 12. This construction directly induces an ensemble of arbitrary mappings with different weights, all learning on the same reward signal. More broadly, this can potentially be combined with any other decomposition scheme, such as the ones we discussed above. For example, in the case of separating negative and positive rewards, one may also deploy two (or more) different mappings for each of the negative and positive reward channels. This certain case results in four channels, two negative and two positive, with proper weights that sum to one.

## 4 ORCHESTRATION OF VALUE-MAPPINGS USING DECOMPOSED REWARDS

### 4.1 ALGORITHM

To have a full orchestration, we next combine value mapping and reward decomposition. We follow the previous steps in Equations 6–10, but now also accounting for the reward decomposition. The core idea here is to replace Equation 6 with 12 and then compute $\widetilde{Q}^{(j)}$ for each reward channel in parallel. In practice, these can be implemented as separate $Q$ tables, separate $Q$ networks, or different network heads with a shared torso. At each time $t$, the algorithm receives all channel outputs $\widetilde{Q}_t^{(j)}$, for $j = 1 \ldots L$, and updates them in accordance with the observed transition. The complete steps are presented in Algorithm 1.

A few points are apropos to remark. Firstly, the steps in the for-loop can be computed in parallel for all $L$ channels. Secondly, as mentioned previously, the mapping function $f_j$ may be different for each channel; however, the discount factor $\gamma$ and both learning rates $\beta_{f,t}$ and $\beta_{reg,t}$ are shared among all the channels and must be the same. Finally, note also that the action $\tilde{a}_{t+1}$, from which all the channels bootstrap, comes from $\arg\max_{a'} Q_t(s_{t+1}, a')$ and not the local value of each channel. This directly implies that each channel-level value $Q^{(j)} = f^{-1}(\widetilde{Q}^j)$ does not solve a channel-level Bellman equation by itself. In other words, $Q^{(j)}$ does not represent any specific semantics such as expected return corresponding to the rewards of that channel. They only become meaningful when they compose back together and rebuild the original value function.

### 4.2 CONVERGENCE

We establish convergence of Algorithm 1 by the following theorem.

---

**Algorithm 1:** Orchestrated Value Mapping.

---

**Input:** (at time $t$)
   $\widetilde{Q}_t^{(j)}$ for $j = 1 \ldots L$
   $s_t, a_t, r_t$, and $s_{t+1}$
**Output:** $\widetilde{Q}_{t+1}^{(j)}$ for $j = 1 \ldots L$

---

**Compute** $r_t^{(j)}$ for $j = 1 \ldots L$

---

**begin**

1    $Q_t(s_t, a_t) := \sum_{j=1}^{L} \lambda_j f_j^{-1} \left( \widetilde{Q}_t^{(j)}(s_t, a_t) \right)$

2    $\tilde{a}_{t+1} := \arg\max_{a'} \left( Q_t(s_{t+1}, a') \right)$

    **for** $j = 1$ *to* $L$ **do**

3      $U_t^{(j)} := r_t^{(j)} + \gamma f_j^{-1} \left( \widetilde{Q}_t^{(j)}(s_{t+1}, \tilde{a}_{t+1}) \right)$

4      $\hat{U}_t^{(j)} := f_j^{-1} \left( \widetilde{Q}_t^{(j)}(s_t, a_t) \right) + \beta_{reg,t} \left( U_t^{(j)} - f_j^{-1} \left( \widetilde{Q}_t^{(j)}(s_t, a_t) \right) \right)$

5      $\widetilde{Q}_{t+1}^{(j)}(s_t, a_t) := \widetilde{Q}_t^{(j)}(s_t, a_t) + \beta_{f,t} \left( f_j \left( \hat{U}_t^{(j)} \right) - \widetilde{Q}_t^{(j)}(s_t, a_t) \right)$

    **end**

**end**

---

**Theorem 1** *Let the reward admit a decomposition as defined by Equation 11, $Q_t(s_t, a_t)$ be defined by Equation 12, and all $\widetilde{Q}_t^{(j)}(s_t, a_t)$ updated according to the steps of Algorithm 1. Assume further that the following hold:*

1. *All $f_j$'s satisfy Assumptions 1 and 2;*

2. *TD error in the regular space (second term in line 4 of Algorithm 1) is bounded for all $j$;*

3. *$\sum_{t=0}^{\infty} \beta_{f,t} \cdot \beta_{reg,t} = \infty$;*

4. *$\sum_{t=0}^{\infty} (\beta_{f,t} \cdot \beta_{reg,t})^2 < \infty$;*

5. *$\beta_{f,t} \cdot \beta_{reg,t} \to 0$ as $t \to \infty$.*

*Then, $Q_t(s, a)$ converges to $Q_t^*(s, a)$ with probability one for all state-action pairs $(s, a)$.*

The proof follows basic results from stochastic approximation theory (Jaakkola et al., 1994; Singh et al., 2000) with important additional steps to show that those results hold under the assumptions of Theorem 1. The full proof is fairly technical and is presented in Appendix A.

We further remark that Theorem 1 only requires the product $\beta_{f,t} \cdot \beta_{reg,t}$ to go to zero. As this product resembles the conventional learning rate in Q-Learning, this assumption is no particular limitation compared to traditional algorithms. We contrast this assumption with the one in the previous proof of Log Q-Learning which separately requires $\beta_{reg}$ to go to zero fast enough. We note that in the case of using function approximation, as in a DQN-like algorithm (Mnih et al., 2015), the update in line 5 of Algorithm 1 should naturally be managed by the used optimizer, while line 4 may be handled manually. This has proved challenging as the convergence properties can be significantly sensitive to learning rates. To get around this problem, van Seijen et al. (2019) decided to keep $\beta_{reg,t}$ at a fixed value in their deep RL experiments, contrary to the theory. Our new condition, however, formally allows $\beta_{reg,t}$ to be set to a constant value as long as $\beta_{f,t}$ properly decays to zero.

A somewhat hidden step in the original proof of Log Q-Learning is that the TD error in the regular space (second term in Equation 9) must always remain bounded. We will make this condition explicit. In practice, with bounds of the reward being known, one can easily find bounds of return in regular as well as $f_j$ spaces, and ensure boundness of $U_t^{(j)} - f_j^{-1}(\widetilde{Q}_t^{(j)})$ by proper clipping. Notably, clipping of Bellman target is also used in the literature to mitigate the value overflow issue (Fatemi et al., 2019). The scenarios covered by Assumption 2, with the new convergence proof due to Theorem 1, may be favorable in many practical cases. Moreover, several prior algorithms such as Q-Learning, Log Q-Learning, and Q-Decomposition can be derived by appropriate construction from Algorithm 1.

### 4.3 REMARKS

#### TIME-DEPENDENT CHANNELS

The proof of Theorem 1 does not directly involve the channel weights $\lambda_j$. However, changing them will impact $Q_t$, which changes the action $\tilde{a}_{t+1}$ in line 2 of Algorithm 1. In the case that $\lambda_j$'s vary with time, if they all converge to their final fixed value soon enough before the learning rates become too small, and if additionally all state-action pairs are still visited frequently enough after $\lambda_j$'s are settled to their final values, then the algorithm should still converge to optimality. Of course, this analysis is far from formal; nevertheless, we can still strongly conjecture that an adaptive case where the channel weights vary with time should be possible to design.

#### SLOPE OF MAPPINGS

Assumption 2 asserts that the derivative of $f_j$ must be bounded from both below and above. While this condition is sufficient for the proof of Theorem 1, we can probe its impact further. The proof basically demonstrates a bounded error term, which ultimately converges to zero under the conditions of Theorem 1. However, the bound on this error term (see Lemma 2 in Appendix A) is scaled by $\delta_{max} = \max_j \delta^{(j)}$, with $\delta^{(j)}$ being defined as

$$\delta^{(j)} = \delta_2^{(j)} / \delta_1^{(j)} - 1, \tag{15}$$

where $\delta_1^{(j)}$ and $\delta_2^{(j)}$ are defined according to Assumption 2 ($0 < \delta_1^{(j)} \leq |f_j'(x)| \leq \delta_2^{(j)}$). In the case of $f_j$ being a straight line $\delta^{(j)} = 0$, thus no error is incurred and the algorithm shrinks to Q-Learning. An important extreme case is when $\delta_1^{(j)}$ is too small while $\delta_2^{(j)}$ is not close to $\delta_1^{(j)}$. It then follows from Equation 15 that the error can be significantly large and the algorithm may need a long time to converge. This can also be examined by observing that if the return is near the areas where $f_j'$ is very small, the return may be too *compressed* when mapped. Consequently, the agent becomes insensitive to the change of return in such areas. This problem can be even more significant in deep RL due to more complex optimization processes and nonlinear approximations. The bottom-line is that the mapping functions should be carefully selected in light of Equation 15 to avoid extremely large errors while still having desired slopes to magnify or suppress the returns when needed. This analysis also explains why logarithmic mappings of the form $f(x) = c \cdot \log(x + d)$ (as investigated in the context of Log Q-Learning by van Seijen et al. (2019)) present unfavorable results in dense reward scenarios; e.g. in the Atari 2600 game of Skiing where there is a reward at every step. In this expression $c$ is a mapping hyperparameter that scales values in the logarithmic space and $d$ is a small positive scalar to ensure bounded derivatives, where the functional form of the derivative is given by $f'(x) = \frac{c}{x+d}$. Hence, $\delta_2 = \frac{c}{d}$, whereas $\delta_1$ can be very close to zero depending on the maximum return. As a result, when learning on a task which often faces large returns, Log Q-Learning operates mostly on areas of $f$ where the slope is small and, as such, it can incur significant error compared to standard Q-Learning. See Appendix B for a detailed illustration of this issue, and Appendix D for the full list of reward density variations across a suite of 55 Atari 2600 games.

## 5 EXPERIMENTAL RESULTS

In this section, we illustrate the simplicity and utility of instantiating new learning methods based on our theory. Since our framework provides a very broad algorithm class with numerous possibilities, deep and meaningful investigations of specific instances go far beyond the scope of this paper (or any single conference paper). Nevertheless, as an authentic illustration, we consider the LogDQN algorithm (van Seijen et al., 2019) and propose an altered mapping function. As discussed above, the logarithmic mapping in LogDQN suffers from a too-small slope when encountering large returns. We lift this undesirable property while keeping the desired magnification property around zero. Specifically, we substitute the logarithmic mapping with a piecewise function that at the break-point $x = 1 - d$ switches from a logarithmic mapping to a straight line with slope $c$ (i.e. the same slope as $c \cdot \log(x + d)$ at $x = 1 - d$):

$$f(x) := \begin{cases} c \cdot \log(x + d) & \text{if } x \leq 1 - d \\ c \cdot (x - 1 + d) & \text{if } x > 1 - d \end{cases}$$

We call the resulting method LogLinDQN, or *Logarithmic-Linear* DQN. Remark that choosing $x = 1 - d$ as the break-point has the benefit of using only a single hyperparameter $c$ to determine both the scaling of the logarithmic function and the slope of the linear function, which otherwise would require an additional hyperparameter. Also note that the new mapping satisfies Assumptions 1 and 2. We then use two reward channels for non-negative and negative rewards, as discussed in the first example of Section 3 (see Equation 13), and use the same mapping function for both channels.

Our implementation of LogLinDQN is based on Dopamine (Castro et al., 2018) and closely matches that of LogDQN, with the only difference being in the mapping function specification. Notably, our LogLin mapping hyperparameters are realized using the same values as those of LogDQN; i.e. $c = 0.5$ and $d \approx 0.02$. We test this method in the Atari 2600 games of the Arcade Learning Environment (ALE) (Bellemare et al., 2013) and compare its performance primarily against LogDQN and DQN (Mnih et al., 2015), denoted by "Lin" or "(Lin)DQN" to highlight that it corresponds to a linear mapping function with slope one. We also include two other major baselines for reference: C51 (Bellemare et al., 2017) and Rainbow (Hessel et al., 2018). Our tests are conducted on a stochastic version of Atari 2600 using *sticky actions* (Machado et al., 2018) and follow a unified evaluation protocol and codebase via the Dopamine framework (Castro et al., 2018).

Figure 1 shows the relative human-normalized score of LogLinDQN w.r.t. the worst and best of LogDQN and DQN for each game. These results suggest that LogLinDQN reasonably unifies the good properties of linear and logarithmic mappings (i.e. handling dense or sparse reward distributions respectively), thereby enabling it to improve upon the per-game *worst* of LogDQN and DQN

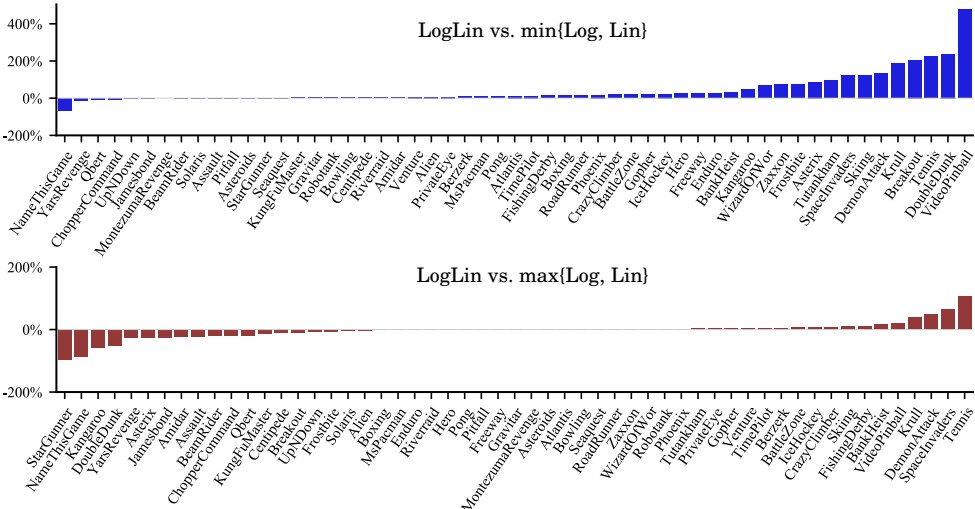

Figure 1: Difference in human-normalized score for 55 Atari 2600 games, LogLinDQN versus the worst (top) and best (bottom) of LogDQN and (Lin)DQN. Positive % means LogLinDQN outperforms the per-game respective baseline.

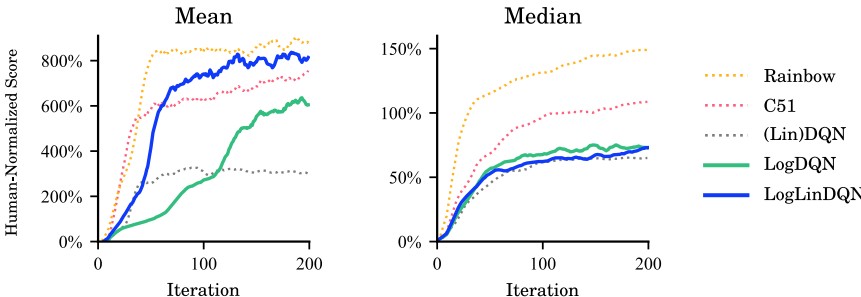

Figure 2: Human-normalized mean (left) and median (right) scores across 55 Atari 2600 games.

(top panel) and perform competitively against the per-game *best* of the two (bottom panel) across a large set of games. Figure 2 shows median and mean human-normalized scores across a suite of 55 Atari 2600 games. Our LogLinDQN agent demonstrates a significant improvement over most baselines and is competitive with Rainbow in terms of mean performance. This is somewhat remarkable provided the relative simplicity of LogLinDQN, especially, w.r.t. Rainbow which combines several other advances including distributional learning, prioritized experience replay, and n-step learning.

## 6 CONCLUSION

In this paper we introduced a convergent class of algorithms based on the composition of two distinct foundations: (1) mapping value estimates to a different space using arbitrary functions from a broad class, and (2) linearly decomposing the reward signal into multiple channels. Together, this new family of algorithms enables learning the value function in a collection of different spaces where the learning process can potentially be easier or more efficient than the original return space. Additionally, the introduced methodology incorporates various versions of ensemble learning in terms of linear decomposition of the reward. We presented a generic proof, which also relaxes certain limitations in previous proofs. We also remark that several known algorithms in classic and recent literature can be seen as special cases of the present algorithm class. Finally, we contemplate research on numerous special instances as future work, following our theoretical foundation. Also, we believe that studying the combination of our general value mapping ideas with value decomposition (Tavakoli et al., 2021), instead of the the reward decomposition paradigm studied in this paper, could prove to be a fruitful direction for future research.

REPRODUCIBILITY STATEMENT

We release a generic codebase, built upon the Dopamine framework (Castro et al., 2018), with the option of using arbitrary compositions of mapping functions and reward decomposition schemes as *easy-to-code* modules. This enables the community to easily explore the design space that our theory opens up and investigate new convergent families of algorithms. This also allows to reproduce the results of this paper through an accompanying configuration script. The source code can be accessed at: https://github.com/microsoft/orchestrated-value-mapping.

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

# A  PROOF OF THEOREM 1

We use a basic convergence result from stochastic approximation theory. In particular, we invoke the following lemma, which has appeared and proved in various classic texts; see, e.g., Theorem 1 in Jaakkola et al. (1994) or Lemma 1 in Singh et al. (2000).

---

**Lemma 1** *Consider an algorithm of the following form:*

$$\Delta_{t+1}(x) := (1 - \alpha_t)\Delta_t(x) + \alpha_t \psi_t(x), \tag{16}$$

*with $x$ being the state variable (or vector of variables), and $\alpha_t$ and $\psi_t$ denoting respectively the learning rate and the update at time $t$. Then, $\Delta_t$ converges to zero w.p. (with probability) one as $t \to \infty$ under the following assumptions:*

1. *The state space is finite;*

2. *$\sum_t \alpha_t = \infty$ and $\sum_t \alpha_t^2 < \infty$;*

3. *$\|\mathbb{E}\{\psi_t(x) \mid \mathcal{F}_t\}\|_W \le \xi \|\Delta_t(x)\|_W$, with $\xi \in (0, 1)$ and $\|\cdot\|_W$ denoting a weighted max norm;*

4. *$Var\{\psi_t(x) \mid \mathcal{F}_t\} \le C\left(1 + \|\Delta_t(x)\|_W\right)^2$, for some constant $C$;*

*where $\mathcal{F}_t$ is the history of the algorithm until time $t$.*

---

Remark that in applying Lemma 1, the $\Delta_t$ process generally represents the difference between a stochastic process of interest and its optimal value (that is $Q_t$ and $Q_t^*$), and $x$ represents a proper concatenation of states and actions. In particular, it has been shown that Lemma 1 applies to Q-Learning as the TD update of Q-Learning satisfies the lemma's assumptions 3 and 4 (Jaakkola et al., 1994).

We define $Q_t^{(j)}(s, a) := f^{-1}\left(\widetilde{Q}_t^{(j)}(s, a)\right)$, for $j = 1 \dots L$. Hence,

$$Q_t(s, a) = \sum_{j=1}^{L} \lambda_j f_j^{-1}\left(\widetilde{Q}_t^{(j)}(s, a)\right) = \sum_{j=1}^{L} \lambda_j Q_t^{(j)}(s, a). \tag{17}$$

We next establish the following key result, which is core to the proof of Theorem 1. The proof is given in the next section.

---

**Lemma 2** *Following Algorithm 1, for each channel $j \in \{1, \dots, L\}$ we have*

$$Q_{t+1}^{(j)}(s_t, a_t) = Q_t^{(j)}(s_t, a_t) + \beta_{reg,t} \cdot \beta_{f,t}\left(U_t^{(j)} - Q_t^{(j)}(s_t, a_t) + e_t^{(j)}\right), \tag{18}$$

*with the error term satisfying the followings:*

1. *Bounded by TD error in the regular space with decaying coefficient*

$$|e_t^{(j)}| \le \beta_{reg,t} \cdot \beta_{f,t} \cdot \delta^{(j)} \left|U_t^{(j)} - Q_t^{(j)}(s_t, a_t)\right|, \tag{19}$$

   *where $\delta^{(j)} = \delta_2^{(j)} / \delta_1^{(j)} - 1$ is a positive constant;*

2. *For a given $f_j$, $e_t^{(j)}$ does not change sign for all $t$ (it is either always non-positive or always non-negative);*

3. *$e_t^{(j)}$ is fully measurable given the variables defined at time $t$.*

---

From Lemma 2, it follows that for each channel:

$$Q_{t+1}^{(j)}(s_t, a_t) = Q_t^{(j)}(s_t, a_t) + \beta_{reg,t} \cdot \beta_{f,t}\left(U_t^{(j)} - Q_t^{(j)}(s_t, a_t) + e_t^{(j)}\right), \tag{20}$$

with $e_t^{(j)}$ converging to zero w.p. one under condition 4 of the theorem, and $U_t^{(j)}$ defined as:

$$U_t^{(j)} := r_t^{(j)} + \gamma\, Q_t^{(j)}(s_{t+1}, \tilde{a}_{t+1}).$$

Multiplying both sides of Equation 20 by $\lambda_j$ and taking the summation, we write:

$$\sum_{j=1}^{L} \lambda_j Q_{t+1}^{(j)}(s_t, a_t) = \sum_{j=1}^{L} \lambda_j Q_t^{(j)}(s_t, a_t) + \beta_{reg,t} \cdot \beta_{f,t} \sum_{j=1}^{L} \lambda_j \left( U_t^{(j)} - Q_t^{(j)}(s_t, a_t) + e_t^{(j)} \right).$$

Hence, using Equation 17 we have:

$$\begin{aligned}
Q_{t+1}(s_t, a_t) &= Q_t(s_t, a_t) + \beta_{reg,t} \cdot \beta_{f,t} \sum_{j=1}^{L} \lambda_j \left( U_t^{(j)} - Q_t^{(j)}(s_t, a_t) + e_t^{(j)} \right) \\
&= Q_t(s_t, a_t) + \beta_{reg,t} \cdot \beta_{f,t} \sum_{j=1}^{L} \lambda_j \left( r_t^{(j)} + \gamma\, Q_t^{(j)}(s_{t+1}, \tilde{a}_{t+1}) - Q_t^{(j)}(s_t, a_t) + e_t^{(j)} \right) \\
&= Q_t(s_t, a_t) + \beta_{reg,t} \cdot \beta_{f,t} \left( r_t + \gamma\, Q_t(s_{t+1}, \tilde{a}_{t+1}) - Q_t(s_t, a_t) + \sum_{j=1}^{L} \lambda_j e_t^{(j)} \right).
\end{aligned}$$
$$(21)$$

Definition of $\tilde{a}_{t+1}$ deduces that

$$Q_t(s_{t+1}, \tilde{a}_{t+1}) = Q_t \left( s_{t+1}, \arg\max_{a'} Q_t(s_{t+1}, a') \right) = \max_{a'} Q_t(s_{t+1}, a').$$

By defining $e_t := \sum_{j=1}^{L} \lambda_j e_t^{(j)}$, we rewire Equation 21 as the following:

$$Q_{t+1}(s_t, a_t) = Q_t(s_t, a_t) + \beta_{reg,t} \cdot \beta_{f,t} \left( r_t + \gamma\, \max_{a'} Q_t(s_{t+1}, a') - Q_t(s_t, a_t) + e_t \right). \quad (22)$$

This is a noisy Q-Learning algorithm with the noise term decaying to zero at a quadratic rate w.r.t. the learning rate's decay; more precisely, in the form of $(\beta_{reg,t} \cdot \beta_{f,t})^2$.

Lemma 1 requires the entire update to be properly bounded (as stated in its assumptions 3 and 4). It has been known from the proof of Q-Learning (Jaakkola et al., 1994) that TD error satisfies these conditions, i.e. $r_t + \gamma \max_{a'} Q_t(s_{t+1}, a') - Q_t(s_t, a_t)$ satisfies assumptions 3 and 4 of Lemma 1.

To prove convergence of mapped Q-Learning, we therefore require to show that $|e_t|$ also satisfies a similar property; namely, not only it disappears in the limit, but also it does not interfere intractably with the learning process during training. To this end, we next show that as the learning continues, $|e_t|$ is indeed bounded by a value that can be arbitrarily smaller than the TD error. Consequently, as TD error satisfies assumptions 3 and 4 of Lemma 1, so does $|e_t|$, and so does their sum.

Let $\delta_{max} = \max_j \delta^{(j)}$, with $\delta^{(j)}$ defined in Lemma 2. Multiplying both sides of Equation 19 by $\lambda_j$ and taking the summation over $j$, it yields:

$$\begin{aligned}
|e_t| = \left| \sum_{j=1}^{L} \lambda_j e_t^{(j)} \right| &\leq \sum_{j=1}^{L} \left| \lambda_j e_t^{(j)} \right| \\
&\leq \sum_{j=1}^{L} |\lambda_j| \cdot \beta_{f,t} \cdot \beta_{reg,t} \cdot \delta^{(j)} \left| U_t^{(j)} - Q_t^{(j)}(s_t, a_t) \right| \\
&\leq \beta_{f,t} \cdot \beta_{reg,t} \cdot \delta_{max} \sum_{j=1}^{L} |\lambda_j| \cdot \left| U_t^{(j)} - Q_t^{(j)}(s_t, a_t) \right| \\
&= \beta_{f,t} \cdot \beta_{reg,t} \cdot \delta_{max} \sum_{j=1}^{L} |\lambda_j| \cdot \left| r_t^{(j)} + \gamma\, Q_t^{(j)}(s_{t+1}, \tilde{a}_{t+1}) - Q_t^{(j)}(s_t, a_t) \right|.
\end{aligned}$$
$$(23)$$

The second line follows from Lemma 2.

If TD error in the regular space is bounded, then $\left| r_t^{(j)} + \gamma\, Q_t^{(j)}(s_{t+1}, \tilde{a}_{t+1}) - Q_t^{(j)}(s_t, a_t) \right| \le K^{(j)}$ for some $K^{(j)} \ge 0$. Hence, Equation 23 induces:

$$|e_t| \le \beta_{f,t} \cdot \beta_{reg,t} \cdot \delta_{max} \sum_{j=1}^{L} |\lambda_j| \cdot K^{(j)}$$

$$= \beta_{f,t} \cdot \beta_{reg,t} \cdot \delta_{max} \cdot K, \tag{24}$$

with $K = \sum_{j=1}^{L} |\lambda_j| \cdot K^{(j)} \ge 0$; thus, $|e_t|$ is also bounded for all $t$. As (by assumption) $\beta_{reg,t} \cdot \beta_{f,t}$ converges to zero, we conclude that there exists $T \ge 0$ such that for all $t \ge T$ we have

$$|e_t| \le \xi \left| r_t + \gamma \max_{a'} Q_t(s_{t+1}, a') - Q_t(s_t, a_t) \right|, \tag{25}$$

for any given $\xi \in (0, 1]$.

Hence, not only $|e_t|$ goes to zero w.p. one as $t \to \infty$, but also its magnitude always remains upper-bounded below the size of TD update with any arbitrary margin $\xi$. Since TD update already satisfies assumptions 3 and 4 of Lemma 1, we conclude that with the presence of $e_t$ those assumptions remain satisfied, at least after reaching some time $T$ where Equation 25 holds.

Finally, Lemma 2 also asserts that $e_t$ is measurable given information at time $t$, as required by Lemma 1. Invoking Lemma 1, we can now conclude that the iterative process defined by Algorithm 1 converges to $Q_t^*$ w.p. one.

PROOF OF LEMMA 2

PART 1

Our proof partially builds upon the proof presented by van Seijen et al. (2019). To simplify the notation, we drop $j$ in $f_j$, while we keep $j$ in other places for clarity.

By definition we have $\widetilde{Q}_t^{(j)}(s, a) = f\left( Q_t^{(j)}(s, a) \right)$. Hence, we rewrite Equations 3, 4, and 5 of Algorithm 1 in terms of $Q_t^{(j)}$:

$$U_t^{(j)} = r_t^{(j)} + \gamma Q_t^{(j)}(s_{t+1}, \tilde{a}_{t+1}), \tag{26}$$

$$\hat{U}_t^{(j)} = Q_t^{(j)}(s_t, a_t) + \beta_{reg,t}\left( U_t^{(j)} - Q_t^{(j)}(s_t, a_t) \right), \tag{27}$$

$$f\left( Q_{t+1}^{(j)}(s_t, a_t) \right) = f\left( Q_t^{(j)}(s_t, a_t) \right) + \beta_{f,t}\left( f(\hat{U}_t^{(j)}) - f(Q_t^{(j)}(s_t, a_t)) \right). \tag{28}$$

The first two equations yield:

$$\hat{U}_t^{(j)} = Q_t^{(j)}(s_t, a_t) + \beta_{reg,t}\left( r_t^{(j)} + \gamma\, Q_t^{(j)}(s_{t+1}, \tilde{a}_{t+1}) - Q_t^{(j)}(s_t, a_t) \right). \tag{29}$$

By applying $f^{-1}$ to both sides of Equation 28, we get:

$$Q_{t+1}^{(j)}(s_t, a_t) = f^{-1}\left( f\left( Q_t^{(j)}(s_t, a_t) \right) + \beta_{f,t}\left( f(\hat{U}_t^{(j)}) - f(Q_t^{(j)}(s_t, a_t)) \right) \right), \tag{30}$$

which can be rewritten as:

$$Q_{t+1}^{(j)}(s_t, a_t) = Q_t^{(j)}(s_t, a_t) + \beta_{f,t}\left( \hat{U}_t^{(j)} - Q_t^{(j)}(s_t, a_t) \right) + e_t^{(j)}, \tag{31}$$

where $e_t^{(j)}$ is the error due to averaging in the mapping space instead of in the regular space:

$$e_t^{(j)} := f^{-1}\left( f\left( Q_t^{(j)}(s_t, a_t) \right) + \beta_{f,t}\left( f(\hat{U}_t^{(j)}) - f(Q_t^{(j)}(s_t, a_t)) \right) \right)$$

$$- Q_t^{(j)}(s_t, a_t) - \beta_{f,t}\left( \hat{U}_t^{(j)} - Q_t^{(j)}(s_t, a_t) \right). \tag{32}$$

Table 1: Ordering of $v$ and $w$ for various cases of $f$.

| $f$ | SEMI-CONVEX | SEMI-CONCAVE |
|---|---|---|
| MONOTONICALLY INCREASING | $f(w) \geq f(v)$ ; $w \geq v$ | $f(w) \leq f(v)$ ; $w \leq v$ |
| MONOTONICALLY DECREASING | $f(w) \geq f(v)$ ; $v \geq w$ | $f(w) \leq f(v)$ ; $v \leq w$ |

We next analyze the behavior of $e_t^{(j)}$ under the Theorem 1's assumptions. To simplify, let us introduce the following substitutions:

$$
\begin{aligned}
a &\rightarrow Q_t^{(j)}(s_t, a_t) \\
b &\rightarrow \hat{U}_t^{(j)} \\
v &\rightarrow (1 - \beta_{f,t})\, a + \beta_{f,t}\, b \\
\tilde{w} &\rightarrow (1 - \beta_{f,t}) f(a) + \beta_{f,t} f(b) \\
w &\rightarrow f^{-1}(\tilde{w})
\end{aligned}
$$

The error $e_t^{(j)}$ can be written as

$$
\begin{aligned}
e_t^{(j)} &= f^{-1}\big((1 - \beta_{f,t})f(a) + \beta_{f,t}f(b)\big) - \big((1 - \beta_{f,t})a + \beta_{f,t}b\big) \\
&= f^{-1}(\tilde{w}) - v \\
&= w - v.
\end{aligned}
$$

We remark that both $v$ and $w$ lie between $a$ and $b$. Notably, $e_t^{(j)}$ has a particular structure which we can use to bound $w - v$. See Table 1 for the ordering of $v$ and $w$ for different possibilities of $f$.

We define three lines $g_0(x)$, $g_1(x)$, and $g_2(x)$ such that they all pass through the point $(a, f(a))$. As for their slopes, $g_0(x)$ has the derivative $f'(a)$, and $g_2(x)$ has the derivative $f'(b)$. The function $g_1(x)$ passes through point $(b, f(b))$ as well, giving it derivative $(f(a) - f(b))/(a - b)$. See Figure 3 for all the possible cases. We can see that no matter if $f$ is semi-convex or semi-concave and if it is increasing or decreasing these three lines will sandwich $f$ over the interval $[a, b]$ if $b \geq a$, or similarly over $[b, a]$ if $a \geq b$. Additionally, it is easy to prove that for all $x$ in the interval of $a$ and $b$, either of the following holds:

$$
g_0(x) \geq f(x) \geq g_1(x) \geq g_2(x) \tag{33}
$$

or

$$
g_0(x) \leq f(x) \leq g_1(x) \leq g_2(x). \tag{34}
$$

The first one is equivalent to

$$
g_0^{-1}(y) \leq f^{-1}(y) \leq g_1^{-1}(y) \leq g_2^{-1}(y), \tag{35}
$$

while the second one is equivalent to

$$
g_0^{-1}(y) \geq f^{-1}(y) \geq g_1^{-1}(y) \geq g_2^{-1}(y). \tag{36}
$$

From the definition of $g_1$ it follow that in all the mentioned possibilities of $f$ combined with either of $a \geq b$ or $b \geq a$, we always have $g_1(v) = \tilde{w}$ and $g_1^{-1}(\tilde{w}) = v$. Hence, plugging $\tilde{w}$ in Equation 35 and Equation 36 (and noting that $f^{-1}(\tilde{w}) = w$) deduces

$$
g_0^{-1}(\tilde{w}) \leq w \leq v \leq g_2^{-1}(\tilde{w}) \tag{37}
$$

or

$$
g_0^{-1}(\tilde{w}) \geq w \geq v \geq g_2^{-1}(\tilde{w}). \tag{38}
$$

Either way, regardless of various possibilities for $f$ as well as $a$ and $b$, we conclude that

$$
|e_t^{(j)}| = |v - w| \leq |g_2^{-1}(\tilde{w}) - g_0^{-1}(\tilde{w})|. \tag{39}
$$

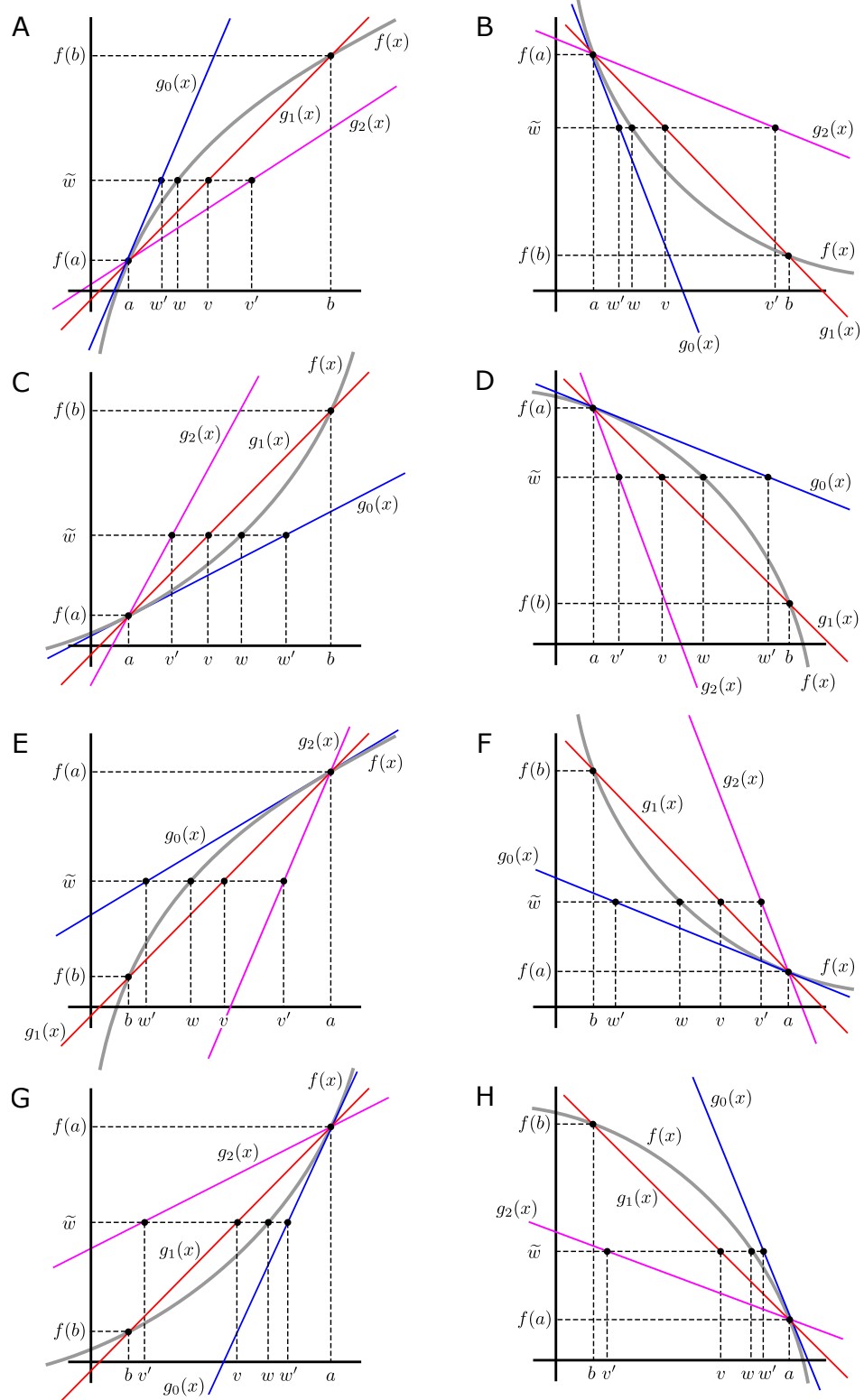

Figure 3: The lines $g_0$, $g_1$, and $g_2$ for all the different possibilities of $f_j$. Cases A–D correspond to $a \leq b$, and cases E–H are the same functions but for $b \leq a$. We use the notations $w' = g_0^{-1}(\widetilde{w})$ and $v' = g_2^{-1}(\widetilde{w})$.

From definition of the lines $g_0$ and $g_2$, we write the line equations as follows:

$$g_0(x) - f(a) = f'(a)(x - a),$$
$$g_2(x) - f(a) = f'(b)(x - a).$$

Applying these equations on the points $(g_0^{-1}(\tilde{w}), \tilde{w})$ and $(g_2^{-1}(\tilde{w}), \tilde{w})$, respectively, it yields:

$$\tilde{w} - f(a) = f'(a)(g_0^{-1}(\tilde{w}) - a),$$
$$\tilde{w} - f(a) = f'(b)(g_2^{-1}(\tilde{w}) - a),$$

which deduce

$$g_0^{-1}(\tilde{w}) = \frac{\tilde{w} - f(a)}{f'(a)} + a \quad ; \quad g_2^{-1}(\tilde{w}) = \frac{\tilde{w} - f(a)}{f'(b)} + a . \tag{40}$$

Plugging the above in Equation 39, it follows:

$$|e_t^{(j)}| = |v - w| \le \left| \frac{\tilde{w} - f(a)}{f'(b)} - \frac{\tilde{w} - f(a)}{f'(a)} \right|$$
$$= \left| \left( \frac{1}{f'(b)} - \frac{1}{f'(a)} \right) (\tilde{w} - f(a)) \right|$$
$$= \left| \left( \frac{1}{f'(b)} - \frac{1}{f'(a)} \right) \Big( (1 - \beta_{f,t})f(a) + \beta_{f,t}f(b) - f(a) \Big) \right|$$
$$= \left| \beta_{f,t} \left( \frac{1}{f'(b)} - \frac{1}{f'(a)} \right) \Big( f(b) - f(a) \Big) \right| . \tag{41}$$

We next invoke the *mean value theorem*, which states that if $f$ is a continuous function on the closed interval $[a, b]$ and differentiable on the open interval $(a, b)$, then there exists a point $c \in (a, b)$ such that $f(b) - f(a) = f'(c)(b - a)$. Remark that based on Assumption 2, $c$ would satisfy $f'_j(c) \le \delta_2^{(j)}$, also that $\frac{1}{f'(b)} - \frac{1}{f'(a)} \le \frac{1}{\delta_1^{(j)}} - \frac{1}{\delta_2^{(j)}}$. Hence,

$$|e_t^{(j)}| \le \left| \beta_{f,t} \left( \frac{1}{f'(b)} - \frac{1}{f'(a)} \right) \Big( f(b) - f(a) \Big) \right|$$
$$= \left| \beta_{f,t} \left( \frac{1}{f'(b)} - \frac{1}{f'(a)} \right) f'(c)(b - a) \right|$$
$$\le \left| \beta_{f,t} \left( \frac{1}{\delta_1^{(j)}} - \frac{1}{\delta_2^{(j)}} \right) \cdot \delta_2^{(j)} \cdot (b - a) \right|$$
$$= \left| \beta_{f,t} \left( \frac{1}{\delta_1^{(j)}} - \frac{1}{\delta_2^{(j)}} \right) \cdot \delta_2^{(j)} \cdot \Big( \hat{U}_t^{(j)} - Q_t^{(j)}(s_t, a_t) \Big) \right| . \tag{42}$$

From Equation 27, it follows that

$$\hat{U}_t^{(j)} - Q_t^{(j)}(s_t, a_t) = \beta_{reg,t} \Big( U_t^{(j)} - Q_t^{(j)}(s_t, a_t) \Big).$$

We therefore can write

$$|e_t^{(j)}| \le \left| \beta_{f,t} \left( \frac{1}{\delta_1^{(j)}} - \frac{1}{\delta_2^{(j)}} \right) \delta_2^{(j)} \Big( \hat{U}_t^{(j)} - Q_t^{(j)}(s_t, a_t) \Big) \right|$$
$$= \left| \beta_{f,t} \left( \frac{1}{\delta_1^{(j)}} - \frac{1}{\delta_2^{(j)}} \right) \delta_2^{(j)} \cdot \beta_{reg,t} \Big( U_t^{(j)} - Q_t^{(j)}(s_t, a_t) \Big) \right|$$
$$= \beta_{f,t} \cdot \beta_{reg,t} \cdot \delta^{(j)} \left| U_t^{(j)} - Q_t^{(j)}(s_t, a_t) \right| ,$$

where $\delta^{(j)} = \delta_2^{(j)} \left( \frac{1}{\delta_1^{(j)}} - \frac{1}{\delta_2^{(j)}} \right)$ is a positive constant. This completes the proof for the first part of the lemma.

PART 2

For this part, it can be directly seen from Figure 3 that for a given $f_j$, the order of $w'$, $w$, $v$, and $v'$ is fixed, regardless of whether $a \geq b$ or $b \geq a$ (in Figure 3, compare each plot A, B, C, and D with their counterparts at the bottom). Hence, the sign of $e_t^{(j)} = w - v$ will not change for a fixed mapping.

PART 3

Finally, we note that by its definition, $e_t^{(j)}$ comprises quantities that are all defined at time $t$. Hence, it is fully measurable at time $t$, and this completes the proof.

## B    DISCUSSION ON LOG VERSUS LOGLIN

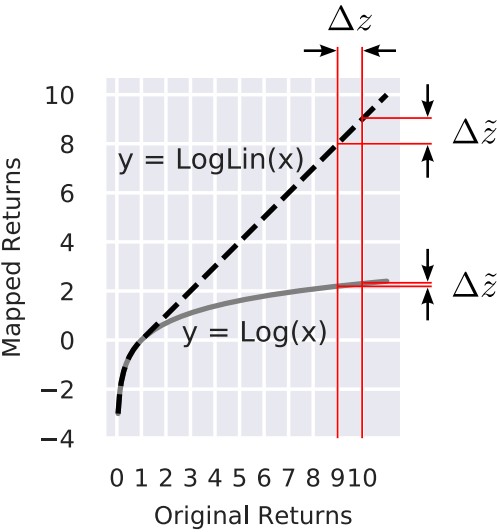

Figure 4: The functions Log and LogLin are illustrated. Remark the compression property of Log as compared to LogLin for returns larger than one.

As discussed in Section 4.3 (Slope of Mappings), the logarithmic (Log) function suffers from a too-low slope when the return is even moderately large. Figure 4 visualizes the impact more vividly. The logarithmic-linear (LogLin) function lifts this disadvantage by switching to a linear (Lin) function for such returns. For example, if the return changes by a unit of reward from 19 to 20, then the change will be seen as 0.05 in the Log space (i.e. $\log(20) - \log(19)$) versus 1.0 in the LogLin space; that is, Log compresses the change by 95% for a return of around 20. As, in general, learning subtle changes is more difficult and requires more training iterations, in such scenarios normal DQN (i.e. Lin function) is expected to outperform LogDQN. On the other hand, when the return is small (such as in sparse reward tasks), LogDQN is expected to outperform DQN. Since LogLin exposes the best of the two worlds of logarithmic and linear spaces (when the return lies in the respective regions), we should expect it to work best if it is to be used as a generic mapping for various games.

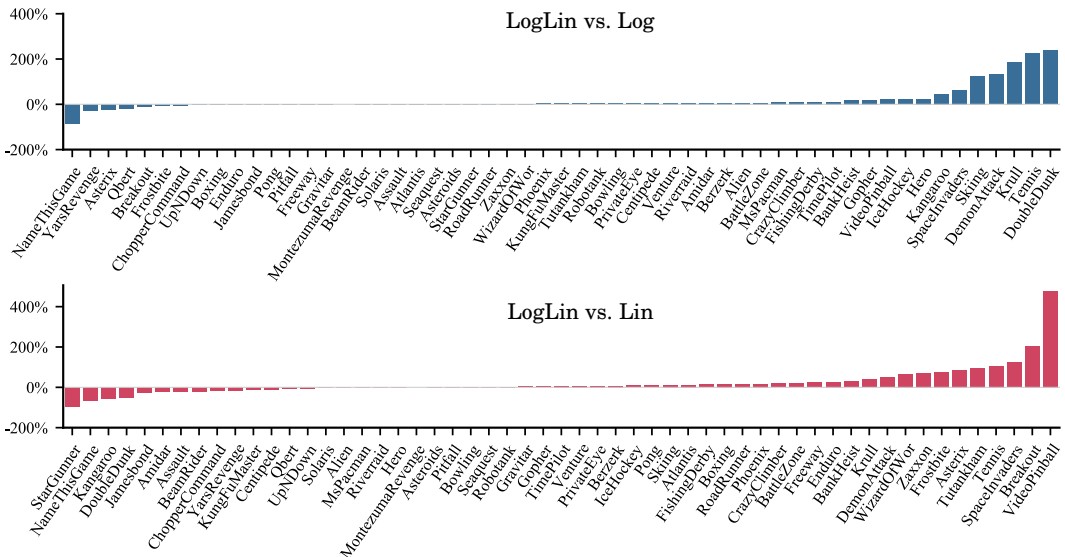

Figure 5: Difference in human-normalized score for 55 Atari 2600 games, LogLinDQN versus LogDQN (top) and (Lin)DQN (bottom). Positive % means LogLinDQN outperforms the respective baseline.

## C  EXPERIMENTAL DETAILS

The human-normalized scores reported in our Atari 2600 experiments are given by the formula (similarly to van Hasselt et al. (2016)):

$$\frac{\text{score}_{\text{agent}} - \text{score}_{\text{random}}}{\text{score}_{\text{human}} - \text{score}_{\text{random}}},$$

where $\text{score}_{\text{agent}}$, $\text{score}_{\text{human}}$, and $\text{score}_{\text{random}}$ are the per-game scores (undiscounted returns) for the given agent, a reference human player, and random agent baseline. We use Table 2 from Wang et al. (2016) to retrieve the human player and random agent scores. The relative human-normalized score of LogLinDQN versus a baseline in each game is given by (similarly to Wang et al. (2016)):

$$\frac{\text{score}_{\text{LogLinDQN}} - \text{score}_{\text{baseline}}}{\max(\text{score}_{\text{baseline}}, \text{score}_{\text{human}}) - \text{score}_{\text{random}}},$$

where $\text{score}_{\text{LogLinDQN}}$ and $\text{score}_{\text{baseline}}$ are computed by averaging over the last 10% of each learning curve (i.e. last 20 iterations).

The reported results are based on three independent trials for LogLinDQN and LogDQN, and five independent trials for DQN.

## D  ADDITIONAL RESULTS

Figure 5 shows the relative human-normalized score of LogLinDQN versus LogDQN (top panel) and versus (Lin)DQN (bottom panel) for each game, across a suite of 55 Atari 2600 games. LogLinDQN significantly outperforms both LogDQN and (Lin)DQN on several games, and is otherwise on par with them (i.e. when LogLinDQN is outperformed by either of LogDQN or (Lin)DQN, the difference is not by a large margin).

Figure 6 shows the raw (i.e. without human-normalization) learning curves across a suite of 55 Atari 2600 games.

Figures 7, 8, and 9 illustrate the change of reward density (measured for positive and negative rewards separately) at three different training points (before training begins, after iteration 5, and after iteration 49) across a suite of 55 Atari 2600 games.

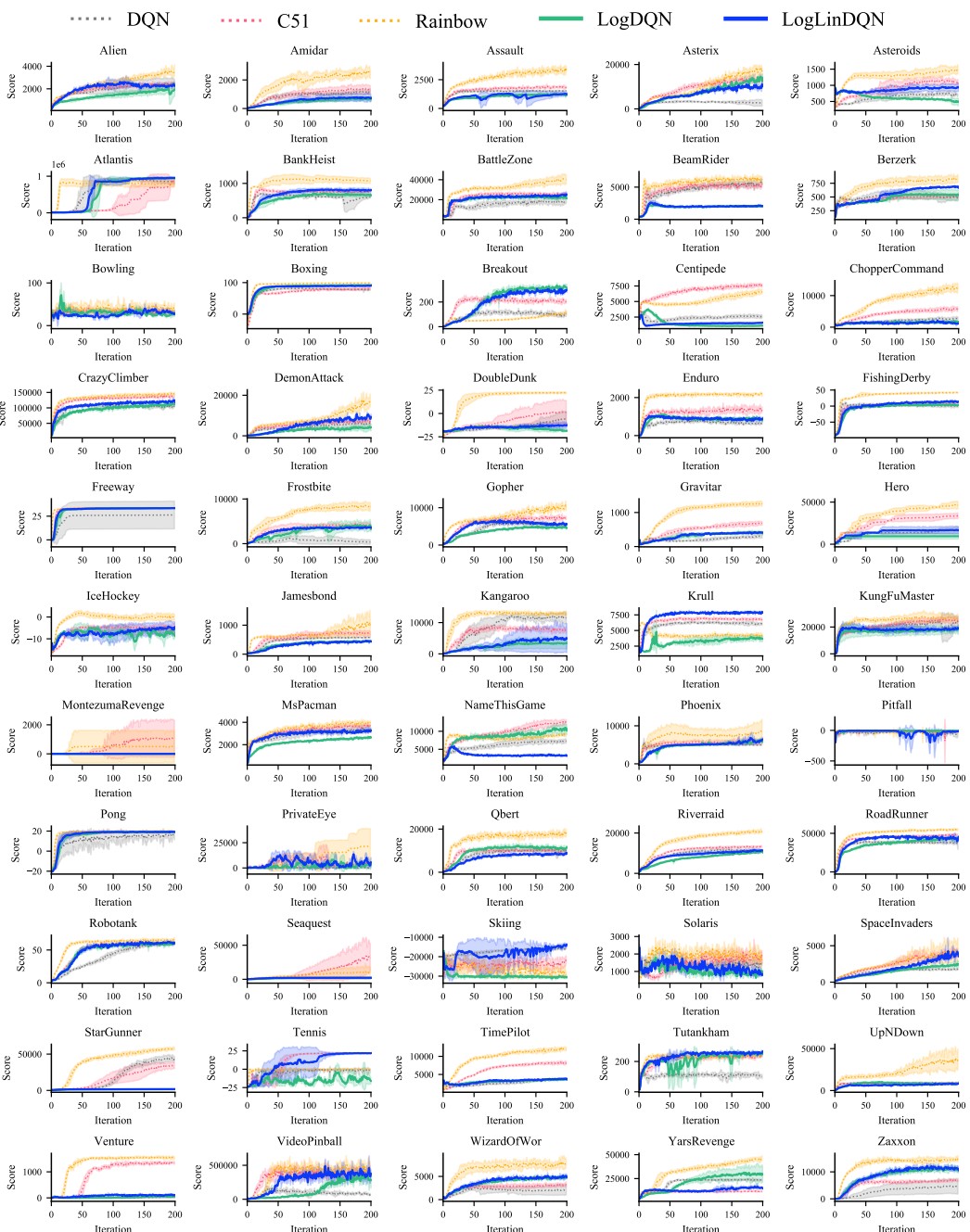

Figure 6: Learning curves in all 55 Atari 2600 games for (Lin)DQN, C51, Rainbow, LogDQN, and LogLinDQN. Shaded regions indicate standard deviation.

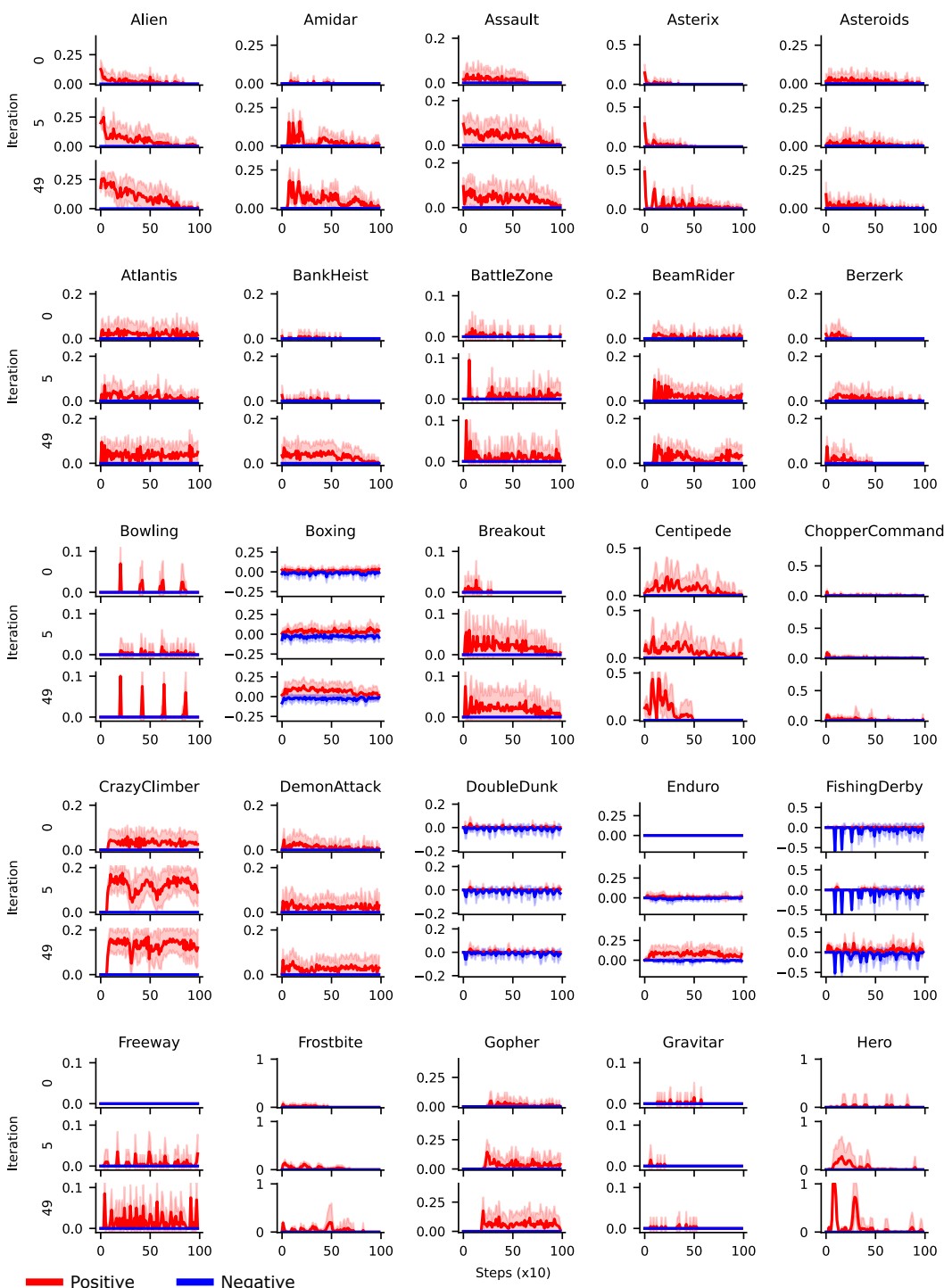

Figure 7: (Part $1/3$) Reward density in the Atari suite. The rewards of each game are measured with three behavioral policies: (i) fully random (iteration 0: the first training iteration), (ii) an $\varepsilon$-greedy policy using $\varepsilon = 0.1$, with state-action values estimated by a (Lin)DQN agent after completing 5 iterations of training, and (iii) similar to (ii) but after 49 training iterations. All experiments are averaged over 20 independent trials (shaded areas depict standard deviation) and for 1000 steps with a moving average window of 10 steps.

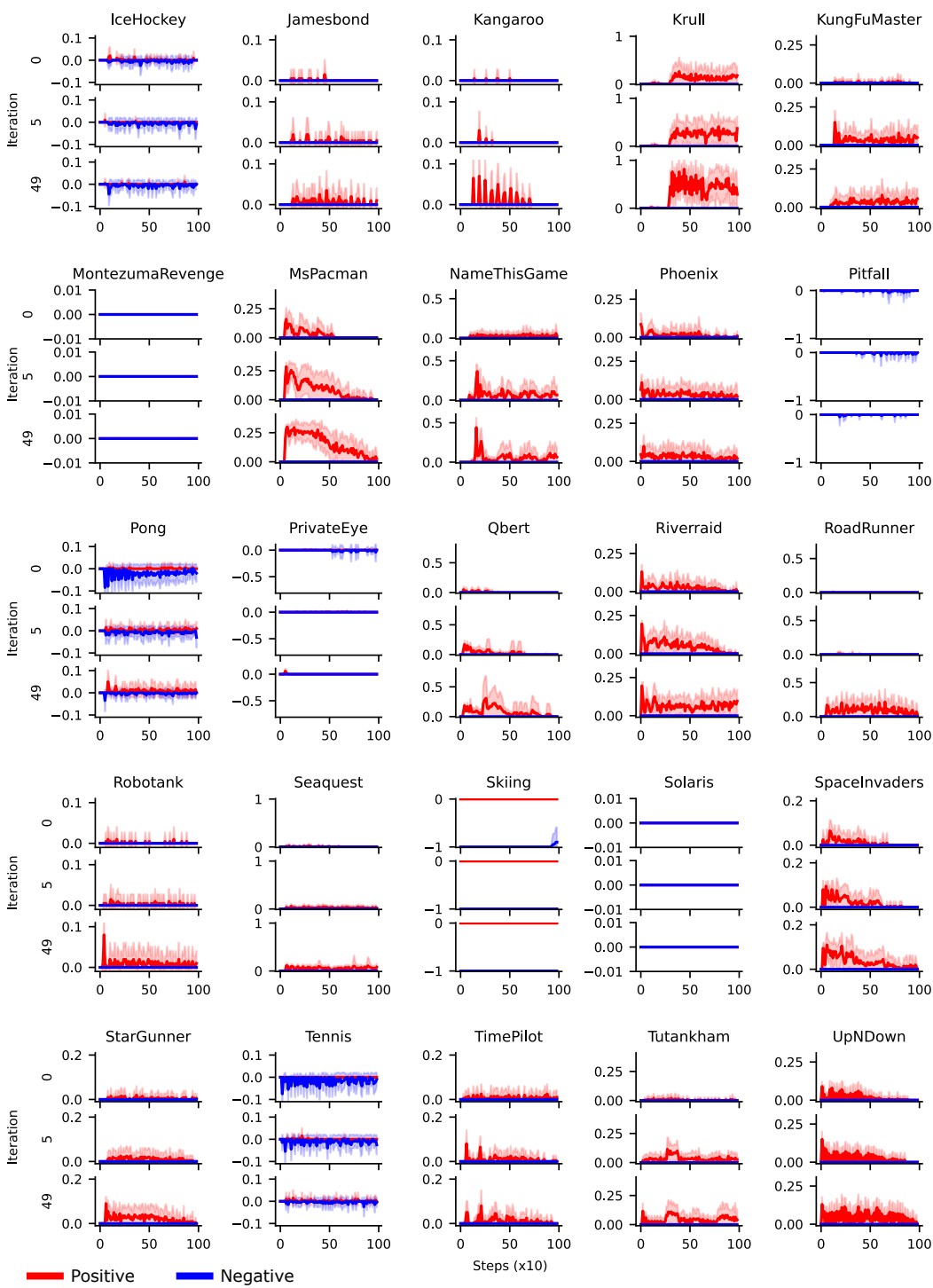

Figure 8: (Part 2/3) Reward density in the Atari suite. The rewards of each game are measured with three behavioral policies: (i) fully random (iteration 0: the first training iteration), (ii) an $\varepsilon$-greedy policy using $\varepsilon = 0.1$, with state-action values estimated by a (Lin)DQN agent after completing 5 iterations of training, and (iii) similar to (ii) but after 49 training iterations. All experiments are averaged over 20 independent trials (shaded areas depict standard deviation) and for 1000 steps with a moving average window of 10 steps.

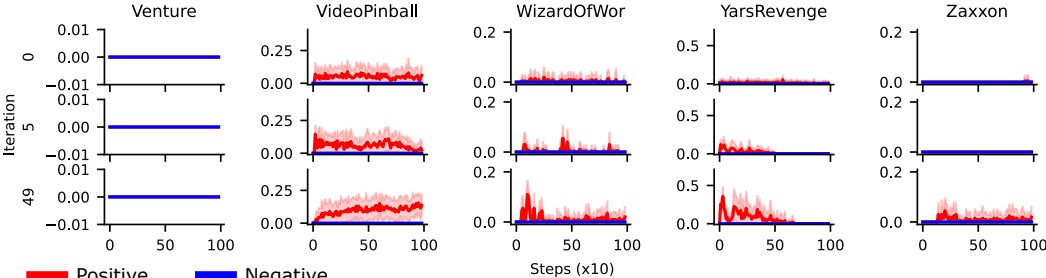

Figure 9: (Part 3/3) Reward density in the Atari suite. The rewards of each game are measured with three behavioral policies: (i) fully random (iteration 0: the first training iteration), (ii) an $\varepsilon$-greedy policy using $\varepsilon = 0.1$, with state-action values estimated by a (Lin)DQN agent after completing 5 iterations of training, and (iii) similar to (ii) but after 49 training iterations. All experiments are averaged over 20 independent trials (shaded areas depict standard deviation) and for 1000 steps with a moving average window of 10 steps.

