# OpenReview forum: "Orchestrated Value Mapping for Reinforcement Learning"
_ICLR.cc/2022/Conference — ICLR 2022 Poster_

### Official Review · Reviewer_159F · 2021-11-02

**Correctness:** 4
**Technical Novelty And Significance:** 3
**Empirical Novelty And Significance:** 3
**Recommendation:** 6
**Confidence:** 3

**Main Review:**

This paper applies the observations from [1] to describe a general algorithm that separates the Q-learning update into two steps: (i) averaging due to environment stochasticity, and (ii)  averaging over different states and actions, moving the latter into the other space. The paper presents these in a clear and simple way, which is very nice. It further formalizes these principles and incorporates them in a generic class of algorithms. It is interesting to see instances of these class in known existing algorithms. The paper is motivated from the view of small agent, big world needing to learn about many things at the same time, which is a promising avenue for autonomous agents. The paper is fairly clear, though it could benefit from some improvements in terms of structure. For instance, the section presenting the first idea describes a proof for a theorem it has not yet been stated, so it is a bit confusing. The authors could maybe move this discussion further down, after the theorem is presented. The assumption of having access to the minimum and maximum return is a bit alarming, how would one have access to those in an \emph{unknown} environment, this seems to imply that the algorithm is not applicable in the same way to all instances of the same problem. The paper feels a bit disconnected, as one idea is presented after the next, and they are presented in a disconnected manner. The ideas are only at the end connected into an algorithm. In the reward decomposition section, there is a mention "SARSA-like update" as common knowledge, without introducing it. The reward decomposition is linear. Why? What are the limitations or assumptions that this is making, and why do we think that channels with separate properties will emerge in the reward space, such that they can be mapped to value functions with different properties, beyond the game-like artificial environments? Where do the values of the weightings applied to the reward channels come from ($\lambda_i$)? The section discussing the "slope of mappings" makes insightful observations, but is a bit hard to follow, and would greatly benefit an illustration. It is only here that the reader is explained why "Assumption 2" was introduced all the way at the beginning, along with a lay word interpretation. The experimental section does not explain the results. What is $d$, where does it come from?

**Summary Of The Paper:**

The paper describes a generic class of algorithms that decompose the reward signal into multiple channels and also map the value function into another generic space via arbitrary functions.  They argue such a class of function is useful to specify certain properties of the learned value on a specific reward signal, also decomposed into channels. They show that known algorithms from the literature are instances of this convergent class of methods.

**Summary Of The Review:**

The paper presents an interesting idea of viewing value decomposition by mapping in a different space after separating the reward functions. Although some parts of the paper still leave open questions, most is clear and very simply written. This framework appears to be useful at least in the Atari domain, but the specifics of the experiment and interpretation of results would benefit some more clarity. The paper contains interesting insights.

---

> ### Author Response · Authors · 2021-11-20
> **Response to Reviewer 159F (1/2)**
>
> Thank you for your constructive and detailed feedback. We are glad that you find the paper insightful. Below we try to address your concerns.
>
> > The section presenting the first idea describes a proof for a theorem it has not yet been stated, so it is a bit confusing. The authors could maybe move this discussion further down after the theorem is presented.
>
> We suppose the reviewer is referring to the last two paragraphs before Section 3. We originally placed it here to give an overview of theoretical issues before our formal results are presented. But we agree that this may be confusing and have moved this part to the section where the proof is presented.
>
> > The assumption of having access to the minimum and maximum return is a bit alarming, how would one have access to those in an *unknown* environment, this seems to imply that the algorithm is not applicable in the same way to all instances of the same problem.
>
> First, we note that our assumption is significantly milder than this. Our generic algorithm only requires the TD error to be bounded that can be satisfied by knowing *upper* and *lower* bounds on the return (which in principle can even be far off compared to the *actual* range of returns). (In practice, knowing or guessing some lower and upper bounds on the per-step rewards is often possible, from which lower and upper bounds on the return can be easily inferred.)
>
> Further, we also remark that even the more limiting assumption of knowing the actual min and max returns is quite common in the literature. The operation of several existing algorithms (including C51, Rainbow, and Implicit Quantile Networks) directly depends on this assumption, where this notion is utilized to determine the support of the distribution (i.e. effectively clipping returns by constructing the support for the distribution of returns).
>
> > In the reward decomposition section, there is a mention of "SARSA-like update" as common knowledge, without introducing it.
>
> This mention of SARSA was purely to discuss a related work, and it is not core to understanding our paper. We feel that given the scope of our mention of SARSA, it doesn't warrant including a description of the algorithm in the paper. However, if the reviewer thinks that adding a reference (e.g. to the original paper and to Sec. 6.4 of Sutton and Barto (2018)) would reduce the burden on the reader, we'd be happy to do so.
>
> > The reward decomposition is linear. Why?
>
> Linear reward decomposition is by construction. This allows for the properties we need to orchestrate various mappings (e.g. handling positive and negative rewards, ensemble learning of values, etc.). As a side benefit, it also helps with the utilization of inductive bias in a broad class of reward possibilities, inline with several other attempts in classical and recent literature as cited in the paper.
>
> > What are the limitations or assumptions that this is making, and why do we think that channels with separate properties will emerge in the reward space, such that they can be mapped to value functions with different properties, beyond the game-like artificial environments?
>
> There are no particular limitations about the reward channels per se, only that the decomposition should be linear with arbitrary weights (as defined in equation (11)). We remark that using *multiple* mappings is a *choice*, not an obligation (a given problem may benefit from only one certain mapping). Importantly, we also highlight that the reason for reward decomposition can solely be enabling certain algorithmic designs, and not *just* to leverage utilization of available inductive biases. We had two examples of the former reason in the paper: (A) handling positive and negative rewards was made possible through reward decomposition in a completely generic manner, and (B) a general way of ensemble Q-learning was made possible again through a similar technique. On the other hand, if the environment can benefit from reward decomposition (some games for example, as you also mentioned) then we can as well have a proper orchestration and benefit from multiple mappings.
>
> Please let us know if this doesn't fully address your question.
>
> > Where do the values of the weightings applied to the reward channels come from ($\lambda_i$)?
>
> It depends. If the reward decomposition is by construction, then $\lambda_i$’s are chosen to satisfy certain properties. For example in the case of handling positive and negative rewards, the weights are by construction, not arbitrary (see equation (13) and the line following it). If the decomposition is to enable handling of different reward sizes/orders, then $\lambda_i$’s should be seen as design parameters to be set for a given task (two examples for a game like Ms. Pacman is given in the paper, see Configurations 1 and 2 at page 5).

---

> > ### Comment · Reviewer_159F · 2021-11-30
> > **Thanks for the response.**
> >
> > Thanks for the response. I will keep the original score, though the idea is interesting, and results promising.

---

> ### Author Response · Authors · 2021-11-20
> **Response to Reviewer 159F (2/2)**
>
> > The section discussing the "slope of mappings" makes insightful observations, but is a bit hard to follow, and would greatly benefit an illustration. It is only here that the reader is explained why "Assumption 2" was introduced all the way at the beginning, along with a lay word interpretation.
>
> Thank you for bringing this up. We have slightly revised the section to make it more clear. Also, we have added a visualization (Figure 4) and a short section (Appendix B) to discuss this further in the special case of Log vs. LogLin mappings.
>
> > The experimental section does not explain the results.
>
> We agree; we had not discussed the results to a good extent. We have now added two new plots in Fig. 1 of Sec. 5 (experimental section), which we believe are quite informative. This is followed by a discussion in the body of Sec. 5 describing how the results of Figs. 1 and 2 should be interpreted. We have also moved the previous plots from this section to Appendix D and discuss them there.
>
> > What is $d$, where does it come from?
>
> Thanks for pointing this out, we had missed including this information in the manuscript.
>
> The parameter $d$ is a small positive value to simply prevent the argument of $\log(\cdot)$ from becoming zero. Moreover, it also prevents the argument to become arbitrarily small; hence, it assures Assumption 2 of the paper. Following the LogDQN implementation by van Seijen et al. (2019), we set $d \approx 0.02$. This choice corresponds to $\gamma^{h}$, where $h=100$ is a generic effective horizon for Atari games. However, we found that this parameter is not too sensitive and other small values may work just as well.
>
> In our LogLinDQN (which has a piecewise log and linear mapping), we wanted the slope of the linear part to be equivalent to $c$ (i.e. the scaling hyperparameter of the log mapping). This is convenient as it only relies on one hyperparameter $c$ to determine the scaling factor of both log and linear parts (as opposed to two hyperparameters in a more general case). To achieve this, the break-point at which the log function is switched to the linear function is $x = 1 - d$ (where $d \approx 0.02$ as explained above).
>
> In the revised version, we now describe the intuition behind $d$ in the *last paragraph of Sec. 4.3 (Slope of Mappings)* and state how that impacts the break-point of the piecewise LogLin function in Sec. 5 (first two paragraphs).

---

### Official Review · Reviewer_1Ecs · 2021-11-02

**Correctness:** 3
**Technical Novelty And Significance:** 3
**Empirical Novelty And Significance:** 2
**Recommendation:** 6
**Confidence:** 4

**Main Review:**

The main contribution of the paper is a framework for mapping multiple reward channels to a "nicer" space in parallel. I find this general concept and framework to be both interesting and novel. I also thought the paper was well presented: the ideas are clearly described and easy to follow. The framework does build upon the work of van Seijen et al (2019), which somewhat hinders the novelty rating, but overall I believe this to be a useful contribution to the literature. In particular, I could see future work that seeks to discover good mappings or pick from a library of potential functions, and this paper provides the platform for that.

Other positives include the softening of theoretical assumptions, which means we can get away with using a fixed learning rate (and have the other handled by Adam etc). This removes the need for yet another hyperparameter and will make implementing this framework (or even the original log Q learning work) easier. While I did not spend too much time on the appendix, I also liked the inclusion of Figure 3 in lemma 2, which made it slightly easier to follow than if it were just writing or mathematical notation.

The only downside for me to this paper is the misalignment between the theory and the experimental results. The theory and discussion talk at length about the advantages of various mappings. This is done with the Pac-Man example, as well as in the section "Slope of Mappings". While these discussions are interesting and highly relevant, the experiment itself only serves to show that the chosen mapping outperforms DQN and log Q learning across the Atari tasks. And while a reason for that performance is provided, this behaviour is never demonstrated experimentally. Put another way, the experiments could conceivably have come from *any* paper that improves upon DQN --- they don't really speak to the power or specifics of the framework here.

I realise that space is an issue, but perhaps the Pacman example can be removed to make more room for an additional experiment? In particular, I could imagine something like a toy domain setup that has a top down view of the reward function and the effect that various choices of mappings have on it, as well as the resulting behaviour when the value functions are learned in the mapped space compared to the original reward function. Obviously, this is a very rough idea, but the main thing I'm looking for here would be to provide empirical support for some of the claims, such as "As a result, when learning on a game which often has a large return, LogDQN operates mostly on areas of $f$ where the slope is small, and it can incur significant error compared to normal DQN."

Minor comments:

1. The second last paragraph in Section 4.2 that talks about boundedness. Do you mean here that if we know $r_{min}$ and $r_{max}$, then we know that the max return is $\dfrac{r_{max}}{1 - \gamma}$ and so we can use this to clip the values if necessary?
2. How are c and d selected in Section 5?
3. Bottom of page 3: "bellow"


**Summary Of The Paper:**

The paper proposes a framework for estimating the Q-value function by decomposing the reward into a linear combination of reward signals or channels. These individual channels are then mapped into a new space, similarly to the log Q-learning approach of van Seijen et al (2019). However, here the framework is extended to a more general class of function (convex, etc). Additionally, some of the theoretical assumptions in the original log Q-learning paper are softened. The method (with a particular instantiation of reward mapping) is tested on the Atari benchmark, and demonstrates improved performance compared to DQN and log Q-learning.


**Summary Of The Review:**

The paper proposes an interesting extension of prior work that offers great potential for expansion in future work. I'm very positive about the framework and theoretical results, but slightly down on the empirical ones, since they do not speak to the particulars of the framework.

---

> ### Author Response · Authors · 2021-11-20
> **Response to Reviewer 1Ecs (1/2)**
>
> Thank you for your constructive and detailed feedback. We are glad that you are very positive about the framework and theoretical results. Below we try to address your concerns.
>
> > And while a reason for that performance [(namely, LogLinDQN outperforming LogDQN and (Lin)DQN)] is provided, this behaviour is never demonstrated experimentally. Put another way, the experiments could conceivably have come from any paper that improves upon DQN --- they don't really speak to the power or specifics of the framework here.
>
> We are not certain that we correctly understand this concern. If the concern is about improvements of LogLinDQN, the following would be our response. However, please do let us know if we have misunderstood the question.
>
> van Seijen et al. (2019) have established that LogDQN outperforms DQN in sparse-reward domains and provided concrete insights on why that is, yet they also observed that LogDQN can underperform DQN in dense-reward domains. One contribution of our paper is also to formally explain the reason for such underperformance (last paragraph of the *Slope of Mapping* subsection) and then how this issue can easily be addressed by the LogLin mapping. Both the issue with Log and the use of this mapping is directly justified by our theory.
>
> Moreover, we hope that our new plots in Figure 1 and the discussion in Appendix B help illustrate why LogLinDQN performs better than LogDQN and (Lin)DQN:
>
> - The updated Figure 1 now shows the relative human-normalized score of *LogLinDQN w.r.t. the **worst and best** of LogDQN and DQN for each game*. These results suggest that LogLinDQN reasonably unifies the good properties of linear and logarithmic mappings (i.e. handling dense or sparse reward distributions respectively), thereby enabling it to improve upon the per-game *worst* of LogDQN and DQN (top panel) and perform competitively against the per-game *best* of the two (bottom panel) across a large set of games.
>
> - Figure 4 of the newly added Appendix B illustrates and discusses the reason for the improvements in more detail.
>
> If this doesn't address your concern, we would be happy to engage further to discuss.
>
> > I realise that space is an issue, but perhaps the Pacman example can be removed to make more room for an additional experiment? In particular, I could imagine something like a toy domain setup that has a top-down view of the reward function and the effect that various choices of mappings have on it, as well as the resulting behaviour when the value functions are learned in the mapped space compared to the original reward function. Obviously, this is a very rough idea, but the main thing I'm looking for here would be to provide empirical support for some of the claims, such as "As a result, when learning on a game which often has a large return, LogDQN operates mostly on areas of $f$ where the slope is small, and it can incur significant error compared to normal DQN."
>
> We thank the reviewer for this suggestion. While we agree that such toy experiments would indeed be valuable, we'd like to emphasize that the goal of this paper is not to prescribe a winning mapping function and orchestration, nor to characterize the properties of specific instantiations. Rather, we wanted to provide the blue-print for creating new classes of convergent algorithms (based on utilization of various mapping functions as well as reward decomposition schemes). We hope you can agree that characterizing the properties of specific mapping functions and orchestrations requires separate detailed study which goes beyond the scope of this paper. We contemplate the probing of special instantiations as (possibly multiple) future work.
>
> > The second last paragraph in Section 4.2 that talks about boundedness. Do you mean here that if we know $r_{\min}$ and $r_{\max}$, then we know that the max return is $\frac{r_{\max}}{1−\gamma}$ and so we can use this to clip the values if necessary?
>
> Yes, and for the min return it would be $\frac{r_{\min}}{1−\gamma}$. For instance, in Atari games, where rewards are clipped to the range [-1, 1], the returns are bounded by $\pm\frac{1}{1−\gamma}$. On a related note, this notion is also utilized in C51 and Rainbow agents to allow them to determine the support of the distribution (i.e. effectively clipping returns by constructing the support for the distribution of returns).
>
> Also, note that we only need some upper and lower bounds of the returns to bound the TD error, which is milder than knowing the *actual* possible min and max returns in the environment.

---

> > ### Comment · Reviewer_1Ecs · 2021-11-27
> > **Response**
> >
> > Dear authors
> >
> > Thank you for the response and updated paper. I really like the figure in appendix B, which adds a lot of intuition. I think the main thing I was looking for were results that you already had (but were maybe not obvious). For example, you take Skiing as a case where you get reward at every step and so log-DQN performs badly. This, coupled with Appendix B and the Skiing plots in Appendix D were essentially what I was interested in seeing.
> >
> > One further idea I had (although I certainly do not expect you to follow through with it) is something like:
> >
> > 1. For each game, record the average magnitude of rewards received during training
> > 2. Then, for log-DQN, plot performance vs 1. and see if there is a correlation between reward magnitudes and performance (as you say, you would expect that games with high rewards do poorly since log compresses these)
> > 3. Repeat for DQN and login DQN.
> >
> > Overall, I still think the paper presents a really nice idea that shows a lot of promise for future work to build upon.

---

> > > ### Author Response · Authors · 2021-11-27
> > > **Response to Reviewer**
> > >
> > > We are glad that you like our addition in Appendix B. We also thank you for the great suggestion to highlight task(s) such as Skiing from our experiments, where because of frequent/dense rewards LogDQN performs poorly while LogLinDQN performs well. We will indeed incorporate this suggestion to complement our discussion in Appendix B.
> > >
> > > Regarding your further idea --- we really like it!
> > > We can measure the *reward density* for a high-performing agent in each game, across a collection of games. Then examine and report the correlations between reward density and the performance of each mapping choice.
> > >
> > > We hope that the reviewer agrees with us that these additions are simple enough to be done for the camera-ready version and, as such, would consider increasing their score to further support our paper.

---

> ### Author Response · Authors · 2021-11-20
> **Response to Reviewer 1Ecs (2/2)**
>
> > How are $c$ and $d$ selected in Section 5?
>
> Thanks for pointing this out, we had missed including this information in the manuscript.
>
> The parameter $d$ is a small positive value to simply prevent the argument of $\log(\cdot)$ from becoming zero. Moreover, it also prevents the argument to become arbitrarily small; hence, it assures Assumption 2 of the paper. Parameter $c$ can be used to jointly control the scaling of the logarithmic segment and the slope of the linear segment (derivative of the linear part is also $c$).
>
> Following the LogDQN implementation by van Seijen et al. (2019), we set $c = 0.5$ and $d \approx 0.02$. The choice of $d$ corresponds to $\gamma^{h}$, where $h=100$ is a generic effective horizon for Atari games (however, we found that this parameter is not too sensitive and other small values may work just as well).
>
> We have now updated the manuscript to describe the intuitions behind these hyperparameters as well as the values that we used for them in our experiments; see the last paragraph of Sec. 4.3 and the first-three paragraphs of Sec. 5.

---

### Official Review · Reviewer_x9kQ · 2021-11-08

**Correctness:** 3
**Technical Novelty And Significance:** 2
**Empirical Novelty And Significance:** 2
**Recommendation:** 6
**Confidence:** 3

**Main Review:**

The strength:

- New general value mapping that generalizes from previous work e.g. Log Q-learning
- The orchestration of value mappings and decomnposed reward that can allow the above general value mapping.
- Theoretical and experimental results to backup the proposed idea

Weakness:

- Contribution on value mapping and reward decomposition can be incremental.
- Experiment results on average are still worse than Rainbow.

In overall, the paper is well written and pursues an interesting research problem. Though the proposed idea of value mapping and reward decomposition is incremental given existing work, it's worth trying and has showed the benefit.

The reward decomposition that is based on fixed and hand-designed configuration will limit the novelty and application. It's too technical comparing to the contributions in Sections 2 and 4.

It will be more useful if there are more choices of $f$. The current evaluation looks limited. Though there is improvement over LogDQN, it's unclear how and where the proposed ideas contribute to the score improvement. More ablation studies will also be helpful.

**Summary Of The Paper:**

This paper proposes a new RL algorithm that contains two principles of value function mapping and reward decomposition. This proposal generalizes many existing RL frameworks such as classical Q-learning, Logarithmic Q-learning, and Q-Decomposition. The paper also provides generic theoretical results to backup the theory. The paper also demonstrates this idea on the suite of Atari 2600 games.

**Summary Of The Review:**

See above
=====================
After rebuttal: The author response has addressed my concerns. I am happy to increase the overall rating.

---

> ### Author Response · Authors · 2021-11-20
> **Response to Reviewer x9kQ (1/2)**
>
> Thank you for your time and feedback. Below we address your concerns.
>
> > Experiment results on average are still worse than Rainbow.
>
> *Rainbow* benefits from various additions (e.g. distributional learning using C51, prioritized experience replay, and n-step learning), none of which is present in our agent. The collective impact of these additions is responsible for making Rainbow a state-of-the-art agent. In fact, *C51 and Rainbow had not been outperformed by a non-distributional agent until very recently* by Vieillard et al. (2020). As such, *LogLinDQN* outperforming C51 and not being so far off w.r.t. Rainbow on the mean performance measure is by itself quite surprising, especially given that our addition is super simple.
>
> - Nino Vieillard, Olivier Pietquin, Matthieu Geist. *Munchausen Reinforcement Learning*. NeurIPS 2020.
>
> > The reward decomposition that is based on fixed and hand-designed configuration will limit the novelty and application. It's too technical comparing to the contributions in Sections 2 and 4.
>
> We emphasize that the reward decomposition is not necessarily hand-designed, though it would be an option when domain knowledge is available. Two important examples are provided in the paper: First, in the case of decomposing rewards into negative and non-negative channels, there is **no** hand-designed decomposition and the decomposition is defined by a fully generic construction. Second, in the ensemble example, the decomposition is defined by using the environment reward for each member of the ensemble (i.e. no decomposition is applied at all).
>
> > It will be more useful if there are more choices of $f$.
>
> The easily conceivable mapping functions that fulfil our convergence conditions are as follows:
>
> 1. Linear
> 2. Logarithmic
> 3. Piecewise Logarithmic and Linear
> 4. Exponential
> 5. Piecewise Exponential and Linear
>
> In our Atari results, we include the performance of **DQN** (which represents **Linear** mapping), **LogDQN** (for **Logarithmic** mapping), and **LogLinDQN** (for **Piecewise Logarithmic and Linear**). As per your suggestion, we are now running DQN with exponential mappings (i.e. cases 4 and 5 above) on a few Atari games. However, these results may not be available before the discussion period as, for a fair comparison, we need to evaluate several hyperparameter settings for exponential mappings.
>
> Nonetheless, the goal of this paper is not to prescribe a winning mapping, rather we wanted to provide the blueprint for creating new convergent algorithms based on various mappings. We hope you agree that investigating the benefits of other mappings (such as exponential or piecewise exponential-linear) requires a separate detailed study that goes beyond the scope of this paper.
>
> Other mapping functions are less intuitive and their construction requires significant work. One potential direction could be to use Normalizing Flows to assert the convergence conditions from our framework to *learn a mapping function that satisfies these conditions*. Such a paradigm, enabled by our theory, allows for finding new mapping functions that are much more complex than the symbolic ones we mentioned above. In this way, *good mappings can be learned on-the-go* during the agent's interaction and training process to benefit the agent in the task(s) at hand. In other words, the agent can learn the mapping function that is most useful for the reward structure of the task, and at different stages of experience.

---

> ### Author Response · Authors · 2021-11-20
> **Response to Reviewer x9kQ (2/2)**
>
> > Though there is an improvement over LogDQN, it's unclear how and where the proposed ideas contribute to the score improvement. More ablation studies will also be helpful.
>
> As discussed in the last paragraph of the *Slope of Mapping* subsection (Section 4.3), the *Log* function suffers from a too-low slope when the return is even moderately large. We added a figure to visualize the impact more vividly (see the newly added Figure 4 of the revised version). The *LogLin* function lifts this disadvantage by switching to a linear function for such returns. For example, if the return changes by a unit of reward from 19 to 20, then the change will be seen as **0.05 in the Log space** ($\log(20) - \log(19)$) versus **1.0 in the LogLin space**; that is, *Log* compresses the change by 95% for a return of around 20. As, in general, learning subtle changes is more difficult and requires more training iterations, in such scenarios normal DQN (i.e. linear mapping) is expected to outperform LogDQN (i.e. logarithmic mapping). On the other hand, when the return is smaller than 1 (such as in sparse-reward tasks), LogDQN is expected to outperform DQN. Since *LogLinDQN exposes the best of the two worlds of log and linear spaces* (when the return lies in the respective regions), we should expect it to work best if it is to be used as a generic mapping for various games.
>
> Thank you for raising this concern. We have now added this discussion in the newly added Appendix B.
>
> Regarding "*more ablation studies*", we believe that DQN and LogDQN (which represent *linear* and *logarithmic* mappings) serve as ablations for LogLinDQN (which has a *logarithmic-linear* mapping). Do you have any other ablations in mind?

---

### Public Comment · ~Rishabh_Agarwal2 · 2021-11-10
**Statistical uncertainty in aggregate scores**

Dear authors,

Their is a sizeable statistical uncertainty in mean/median scores [1] on Atari 2600 games and it should be reported for reliable evaluation. A simple way to do so is to use stratified bootstrap confidence intervals [1] even when using only a handful of runs. You can easily do so using the library at https://github.com/google-research/rliable or the [colab notebook](https://bit.ly/statistical_precipice_colab).

Regarding aggregate metrics, mean could be easily dominated by outliers while median remains unchanged even when  scores on half of the task is set to 0. A better metric might be interquartile mean across all runs and tasks. Additionally, performance profiles would capture variability in performance across tasks and runs and percentiles/ aggregate metrics can be easily read from such profiles.

[1] Agarwal, R., Schwarzer, M., Castro, P.S., Courville, A. and Bellemare, M.G., 2021. Deep reinforcement learning at the edge of the statistical precipice. In NeurIPS.

---

### Author Response · Authors · 2021-11-20
**General Response to Reviewers**

We thank all our reviewers for their time, detailed suggestions and constructive comments. We have uploaded a revised version of our paper containing some clarifications and additions. We invite the reviewers to have a second look and reach out with more questions and comments.

For major updates, see the following in the revised version:

- **Figure 1**: Newly added plots reporting the relative normalized score of **LogLinDQN** (*logarithmic-linear mapping*) versus the worst (top panel) and best (bottom panel) of **LogDQN** (*logarithmic mapping*) and **DQN** (*linear mapping*).

- Last paragraph of **Slope of Mappings** subsection (Section 4.3): An improved discussion of logarithmic mapping, followed by a discussion of its pitfalls w.r.t. linear mapping.

- **Appendix B** and **Figure 4**: Added a detailed discussion and a new illustration of the pitfall of logarithmic mapping versus linear mapping, motivating the case for logarithmic-linear mapping.

- **Section 5**:
   - Added a detailed discussion of the hyperparameters $c$ and $d$, what they correspond to, and the values we use for them in LogLinDQN.
   - Added detailed discussions of the results shown in Figures 1 and 2.

- The original Atari barplots in the submission version are now moved to **Appendix D** together with an added explanation of the results.

---

### Decision · Program_Chairs · 2022-01-20

**Decision:**

Accept (Poster)

**Comment:**

Description of paper content:

The paper provides a framework to develop a family of algorithms that decompose rewards into linear combinations of several reward channels. The value functions per channel are estimated in a new space using an invertible function transformation, f. The framework encompasses several previously published algorithms, including Log Q-Learning. Conditions are provided for acceptable choices of f. Convergence to the optimal Q function in the tabular case is proven for a special learning update.

Summary of paper discussion:

All review scores were above the acceptance threshold. Overall, the reviewers found the idea interesting, the theoretical results satisfying, and the writing and presentation clear. Initial concern about the directedness of the experiments in showing the usefulness of this particular theoretical framework to explain performance improvements was allayed when some of the results in the paper (e.g. reward density in Atari Skiing) were re-emphasized. Generally, all reviewers felt that this was a nice, thorough contribution with the demerit that the framework lacked “a killer application” experimentally.